# Fast hierarchical processing of orthographic and semantic parafoveal information during natural reading

Lijuan Wang [1] ✉, Steven Frisson[1], Yali Pan [1,4] ✉ & Ole Jensen [1,2,3,4]

In reading, information from parafoveal words is extracted before direct fixation; however, it is debated whether this processing is restricted to orthographic features or also encompasses semantics. Moreover, the neuronal mechanisms supporting parafoveal processing remain poorly understood. We co-registered MEG and eye-tracking data in a natural reading paradigm to uncover the timing and brain regions involved in parafoveal processing. Representational similarity analysis revealed that parafoveal orthographic neighbours (e.g., "writer" vs. "waiter") showed higher representational similarity than non-neighbours (e.g., "writer" vs. "police"), emerging ~68 ms after fixation onset on the preceding word (e.g., "clever") in the visual word form area. Similarly, parafoveal semantic neighbours (e.g., "writer" vs. "author") exhibited increased representational similarity at ~137 ms in the left inferior frontal gyrus. Importantly, the degree of orthographic and semantic parafoveal processing was correlated with individual reading speed. Our findings suggest fast hierarchical processing of parafoveal words across distinct brain regions, enhancing reading efficiency.

Reading is a seemingly effortless process that allows us to absorb vast amounts of information quickly. To read efficiently, readers not only process currently fixated words in the fovea but also pre-process upcoming words in the parafovea[1]. Studies have shown that masking parafoveal words can severely impair reading speed[2–4]. This suggests that parafoveal processing is essential for fluent reading, as it allows readers to extract information from the to-be-fixated word, providing a head start and thus reducing processing time when the word is subsequently fixated upon[5,6]. In some cases, if a word has been sufficiently processed in the parafovea, the reader may even skip it altogether[7–10]. In reading research, parafoveal processing has been intensely investigated using eye-tracking[1] and event-/fixation-related potentials (ERPs/FRPs)[11], with foci on the types of information processed and the associated timing. The current study aims to investigate how different levels of information from the same word, specifically orthography and semantics, are organised temporally

(i.e., time course) and spatially (i.e., brain regions) during parafoveal processing.

Eye-tracking studies have provided valuable insights into the types of information that can be extracted from parafoveal words using the boundary paradigm[12]. In this paradigm, an invisible boundary is placed just to the left of the *target word*. While the reader's gaze remains to the left of this boundary, a *preview word* occupies the position of the target word. Once the eyes cross the boundary, the preview word is replaced by the target word. If the preview shares certain linguistic features with the target, such as orthography (e.g., "sweet" and "sleet"), phonology (e.g., "sweet" and "suite"), or semantics (e.g., "sweet" and "sugar"), the reader's fixation duration on the target word is reduced—a phenomenon known as the *preview benefit* (for a review see ref. 1). Preview benefits have been consistently observed for orthographic[13–15] and phonological[16–18] features, supporting parafoveal processing of these aspects. Semantic preview benefits, however,

[1]Centre for Human Brain Health, School of Psychology, University of Birmingham, Birmingham, UK. [2]Department of Experimental Psychology, University of Oxford, Oxford, UK. [3]Oxford Centre for Human Brain Activity, Oxford Centre for Integrative Neuroimaging, Department of Psychiatry, University of Oxford, Oxford, UK. [4]These authors jointly supervised this work: Yali Pan, Ole Jensen. ✉e-mail: lxw203@student.bham.ac.uk; Y.Pan.1@bham.ac.uk

remain debated: earlier studies found no evidence for semantic preview benefits[19–21], while later research suggested that semantic preview benefits occur only under specific conditions[22–26], such as a capitalised initial letter[22] or a constraining context[26] of preview/target word. Other eye-movement studies have investigated parafoveal processing by measuring which characteristics of parafoveal words influence the processing of the currently fixated word, i.e., parafoveal-on-foveal (PoF) effects. While orthographic PoF effects are well established (for a review see ref. [1]), the results from eye movement studies have largely not provided evidence in favour of lexical and semantic PoF effects[27,28]. Taken together, eye-tracking evidence remains inconclusive as to whether semantic features can be extracted from the parafovea.

Electrophysiological studies have provided valuable insights into the time course of orthographic[29–32] and semantic parafoveal processing[29,33–41]. Most of these studies have used passive reading paradigms, such as the flanker paradigm, in which sentences are presented word-by-word at fixation and flanked by parafoveal words[32–38], and ERPs were obtained. Other studies have employed more natural saccadic reading paradigms, co-registering eye-tracking and EEG to obtain FRPs[29–31,39,40]. Early ERP components, such as the P1 and N1, have been shown to reflect orthographic parafoveal processing, with their amplitudes modulated by the orthographic properties of parafoveal words[29,30]. Semantic parafoveal processing, in contrast, has been indexed by the N400 component, with larger amplitudes observed for parafoveal words that are semantically incongruent with the sentence context compared to congruent ones[39,40]. However, the parafoveal N400 effect typically occurs more than 250 ms after fixation onset on the pre-target word by which time the target word is often already in the fovea, given that typical fixation durations during natural reading are ~200 ms. Consequently, the parafoveal N400 effect likely captures a later stage of parafoveal semantic processing. Earlier neural mechanisms reflecting the onset of parafoveal semantic processing may exist[41,42], but the existing evidence remains limited.

Recently, rapid invisible frequency tagging (RIFT) has shown potential for measuring the early onset of parafoveal processing during natural reading[41,43]. This technique involves flickering the location of the parafoveal word at a high frequency, such as 60 Hz. Tagging responses to the visual flicker can be detected in the brain and used to measure the degree of attention allocated to the parafoveal word. Studies have shown that these tagging responses are modulated by the lexical[43] and semantic[41] information of the parafoveal word within 100 ms of fixating the preceding word. Despite providing early-onset timing information, RIFT only identifies the attentional modulation effect of the parafoveal word rather than the information extraction itself. Furthermore, due to the limitation in the brain regions that respond to visual flickers, the parafoveal processing effect can only be observed in the visual cortex, which is well known to be outside the typical language-processing network[44,45]. Therefore, a technique that can directly measure the extracted parafoveal information with reliable spatial localisation is needed.

Representational similarity analysis (RSA) combined with Magnetoencephalography (MEG)[46,47] offers a unique opportunity to directly measure extracted parafoveal information with excellent temporal and spatial sensitivity. RSA is based on the assumption that items sharing similarities in specific aspects produce similarly distributed patterns of neural activity compared with dissimilar items[46]. As such, it can be used to measure multiple levels of representation by manipulating similarities across different aspects of items. When combined with electrophysiological recordings, RSA can reveal when specific types of information are represented in the brain[47]. Moreover, RSA can be applied in the source space of MEG data with a searchlight approach to identify the brain regions producing this representational similarity. Several studies have employed RSA with EEG/MEG data to investigate the time course of pre-activation of semantic information during language comprehension[48–51]. These studies compared the similarity of neural activity patterns in the interval when the same words (within-pairs) versus different words (between-pairs) could be predicted in a passive reading paradigm. In the present study, we embraced a similar RSA approach in a natural reading paradigm.

Our study aims to address two core questions: (1) Can we identify specific representational activity associated with orthographic and semantic information of a parafoveal word before it is fixated? (2) If so, what are the neuronal time courses and brain areas associated with orthographic and semantic parafoveal processing? To investigate these questions, we simultaneously recorded MEG and eye movements in a natural reading paradigm (Fig. 1a). We selected a set of critical words (e.g., "writer"), each paired with an orthographic neighbour (e.g., "waiter") and a semantic neighbour (e.g., "author"). Each critical word and its two neighbours formed a triplet of target words and were embedded in different sentences. Each target triplet was paired with another triplet (e.g., "police/policy/guards"). Importantly, pre-target words were the same (e.g., "clever") within each triplet and between its matched triplet, i.e., all six sentences shared identical pre-target words (Fig. 1b). The RSA analysis focused on the pre-target fixation period–when target words are in the parafovea (Fig. 2). Any difference in representational similarity between orthographically similar target words (e.g., "writer" & "waiter") and unrelated words during this period would indicate parafoveal orthographic processing. Similarly, differences between semantically similar words (e.g., "writer" & "author") and unrelated words would indicate parafoveal semantic processing. Finally, we employed a searchlight method to identify the brain areas supporting orthographic and semantic parafoveal processing. In short, our design and methodology allow us to track whether and when orthographic and semantic information is extracted from the upcoming word, and which brain areas are involved in these processes.

## Results
### The time course of orthographic and semantic parafoveal processing

Using RSA, we first compared the similarity of MEG activity patterns for orthographically similar target words (orthographic within pairs) with those for dissimilar target words (between pairs) during the pre-target fixation period (Fig. 2). MEG data were segmented into epochs of −0.2 to 0.5 s, relative to the onset of the first fixation on pre-target words. For each participant, we computed pairwise correlations between the spatial activation patterns of all MEG sensors for orthographically similar target words (orthographic within pairs, e.g., "writer" and "waiter" in Fig. 1b) and averaged these correlations. We also calculated the average correlation for between pairs (e.g., "writer" and "police"). This process was repeated at every millisecond within the interval, yielding time series for both orthographic within-pair correlations ($R_{orth}$) and between-pair correlations ($R_{between}$). Finally, we calculated the grand average of these time series across all participants.

We found that the similarity in brain activity was higher when the parafoveal target words were orthographically similar, compared to when they were dissimilar (Fig. 3a). This difference was significant in the 68–186 ms interval after the fixation onset on the pre-target words (shaded area in Fig. 3a, cluster-based permutation test: $p < 0.001$; 5000 iterations). We note that the values of the correlations are relatively low, but they are comparable to previously published studies using a similar approach[48–51]. As the statistical testing demonstrates, the differences in correlations are robust across participants. Furthermore, to maintain a consistent definition of orthographic neighbours across pairs, we excluded four orthographic within pairs that differed by two letters rather than one; the parafoveal orthographic effect remained robust (66–189 ms, cluster-based permutation test: $p < 0.001$). These findings provide neuronal evidence for parafoveal orthographic processing emerging at ~68 ms after the fixation onset on the pre-target word.

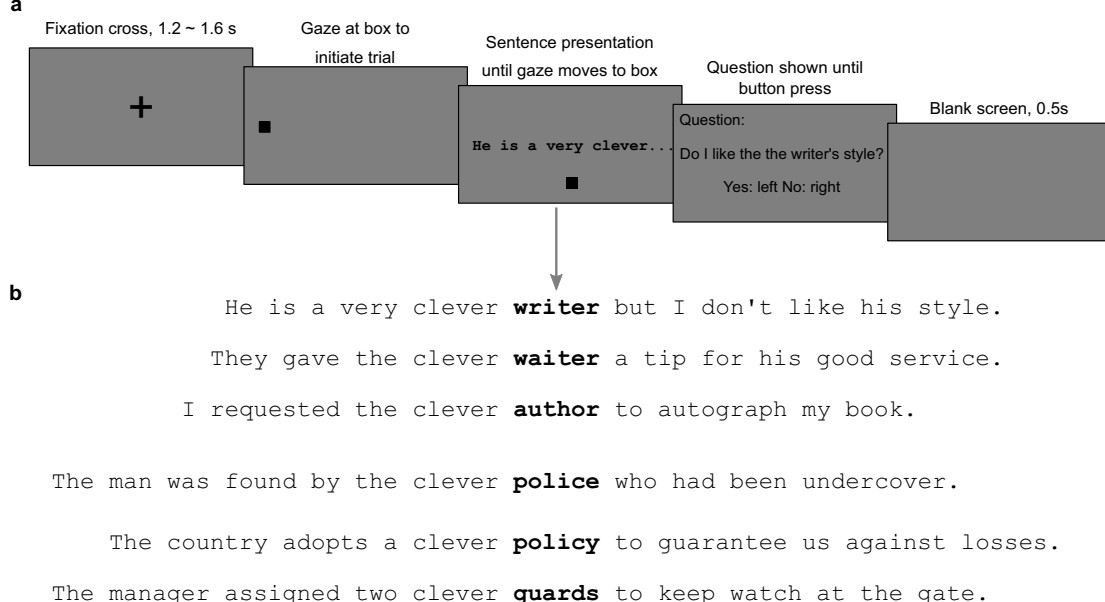

Fig. 1 | Experimental design and example of a sentence sextet. a Experimental design. Participants were instructed to read sentences silently while eye movements and brain activity were recorded simultaneously using an eye tracker and MEG. Each trial started with a 1.2–1.6 s fixation cross in the centre of a grey screen. Then a square was presented on the left side of the screen. The onset of an upcoming sentence was triggered by gazing at the square for at least 0.2 s. After reading the sentence, participants were asked to fixate on a square below the screen for 0.1 s to terminate the trial. Twenty-five percent of the sentences were followed by a simple yes-or-no comprehension question to ensure careful reading. b Example of a sentence sextet. 1. Sentences were constructed in triplets, wherein each sentence included a target word (shown in bold for illustration purposes, not in the actual experiment), which could either be the critical word (e.g., "writer"), its orthographic neighbour (e.g., "waiter"), or its semantic neighbour (e.g., "author"). Each triplet was paired with another triplet, embedding a similar structure of target words (e.g., "police/policy/guards"). Pre-target words were the same (e.g., "clever") within a sentence sextet (a group of six sentences formed by pairing two triplets).

We followed a similar procedure to investigate semantic parafoveal processing. We found that the similarity of MEG activity patterns was higher for semantically similar parafoveal target words compared with dissimilar words (Fig. 3b); this difference was significant in the 137–247 ms after the fixation onset on the pre-target words (shaded area in Fig. 3b, cluster-based permutation test: $p < 0.001$; 5000 iterations). To confirm that the observed semantic effect was not produced by the contextual information preceding the target words, we evaluated the semantic similarity of contexts between the paired sentences preceding the target words using latent semantic analysis (LSA)[52] (See "Methods"). The results showed no difference in the LSA score when comparing the contexts for within-pair and between-pair target words ($t_{(238)} = 0.95$, $p = 0.34$, Cohen's $d = 0.12$, two-sided independent-samples $t$-test), ruling out the possibility that the effect is from the semantic similarity of sentence contexts. These findings provide neuronal evidence demonstrating that parafoveal words are processed at the semantic level at ~137 ms after the onset of fixation on the pre-target word. One potential concern is that the observed semantic effect (137–247 ms) may, in some cases, include early foveal processing of the target word, as short pre-target fixations could allow the eyes to move to the target during this window. To address this, we conducted a control analysis in which we excluded trials where fixation shifted to the target word within 247 ms, the semantic parafoveal effect remained significant ($p = 0.041$; cluster permutation test), again emerging at ~140 ms (see Supplementary Fig. 1), confirming that the effect is not attributable to foveal processing of the target word.

Considering the above neuronal evidence, parafoveal processing is not limited to low-level orthographic information but extends to high-level semantic information. Our results also demonstrated temporally distinct stages in parafoveal processing: low-level orthographic information is available first, and higher-level semantic information is available soon after, and these happen when the pre-target word is under fixation. To understand the dynamics of word representations, we also conducted RSA analyses time-locked to the onset of fixations on the target and post-target words (see Supplementary Fig. 2 for results).

### Parafoveal processing was related to reading speed
To investigate whether the extraction of parafoveal orthographic and semantic information affects reading proficiency, we computed the correlation between these neuronal effects and individual reading speed. We assessed the magnitude of orthographic and semantic parafoveal effects by averaging the differences in $R$-values when significant differences in the RSA analysis emerged (orthography: 68–186 ms; semantics: 137–247 ms after the fixation onset of the pre-target word). The reading speed of each participant was measured as the number of words read per second, calculated by dividing the total number of words across all sentences by the participant's total reading time. Our analysis revealed that both orthographic and semantic parafoveal effects significantly correlated with individual reading speed (orthography: Fig. 3c, R = 0.42, $p = 0.011$; semantics: Fig. 3d, R = 0.34, $p = 0.044$, Spearman correlation), suggesting that individuals with stronger orthographic and semantic parafoveal effects tended to be faster readers. The orthographic and semantic parafoveal effects were not correlated ($R = 0.12$, $p = 0.488$, Spearman correlation), indicating that these two types of parafoveal information extraction contribute independently to reading speed.

### Neuronal sources underlying parafoveal processing
To identify the neuronal sources underlying the observed orthographic and semantic parafoveal processing, we performed RSA with a searchlight method at both sensor and source levels. Sensor-level analysis was conducted separately for magnetometers and gradiometers, using searchlight patches of 20 sensors for magnetometers and 40 sensors for gradiometers (the gradiometers were arranged in pairs of two at each recording location in the Neuromag system). For

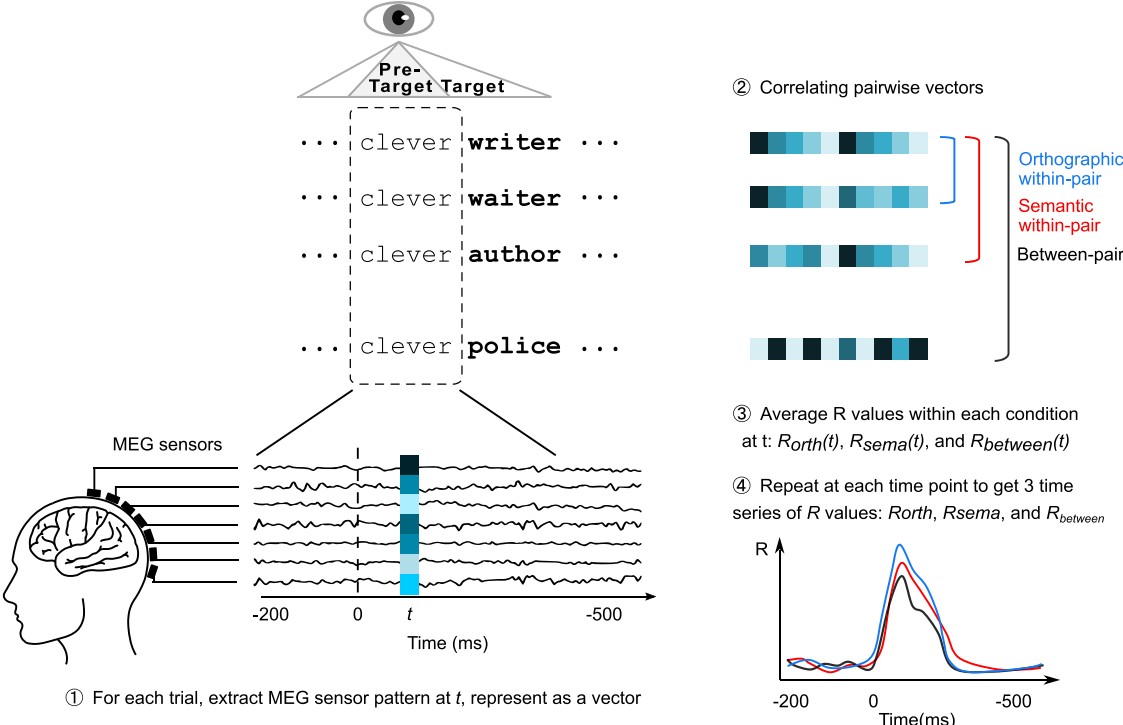

**Fig. 2 | Schematic illustration of our representational similarity analysis.** 1. For each trial, at each time point $t$ (from −200 to 500 ms relative to the fixation onset on the pre-target word, e.g., "clever"), we extracted the MEG signals across sensors to create a vector, representing the brain activity pattern at time $t$. 2. We quantified representational similarity between each pair by computing the Pearson correlation coefficient ($R$) between their corresponding vectors. 3. Then we averaged the $R$-values across all pairs within each condition to obtain the average correlations for three conditions at $t$: $R_{orth}(t)$, $R_{sema}(t)$, and $R_{between}(t)$. 4. We repeated the procedure at every millisecond after the fixation onset on the pre-target; this yielded 3 time series of pairwise correlations: $R_{orth}$, $R_{sema}$, and $R_{between}$. Note that $R_{orth}$, $R_{sema}$, and $R_{between}$ denote the average correlations for orthographic within pairs, semantic within pairs, and unrelated between pairs, respectively.

the orthographic parafoveal processing, we compared the similarity of neural activity patterns between orthographic within pairs and between pairs ($\Delta R_{orth}$) within each searchlight patch, during the interval when the orthographic parafoveal effect was robust (68–186 ms in Fig. 3a). We found that sensors involved in orthographic parafoveal processing spanned occipital, temporal, parietal, and frontal regions, exhibiting a notable left-lateralised distribution (Fig. 4a). This was observed in large clusters identified using a cluster-randomisation approach in both magnetometers ($p < 0.001$) and gradiometers ($p < 0.001$). For the source-level analysis, individual-subject source-reconstructed MEG signals were obtained using a brain surface-constrained minimum norm estimate approach and were converted to a common space (See *Methods* for details). We then applied searchlight RSA to the source data, using a searchlight consisting of 2000 vertices across the cortical surface (out of 20,484 vertices). The vertices showing the maximal difference between orthographic within and between pairs emerged in the left ventral occipitotemporal region (lvOT; Fig. 4b left)—a region overlapping with the visual word-form area (VWFA)[53].

A searchlight RSA approach was also used to identify the neuronal sources of semantic parafoveal processing in the 137–247 ms interval (in Fig. 3b), mirroring the approach used for orthographic parafoveal processing. The topography of sensor-level data using magnetometers (Fig. 4c, left) showed that the greatest difference in representational similarity between semantic within pairs and between pairs ($\Delta R_{sema}$) was produced over frontal and left temporal regions ($p = 0.04$). The topography of sensor-level data using gradiometers (Fig. 4c, right) showed that the greatest $R$-values difference was over frontal regions ($p = .008$). A source-level searchlight approach revealed that the neuronal generator of the semantic parafoveal effect was localised at the left inferior frontal gyrus (LIFG) (Fig. 4d left)—a core region of the language network.

We thus propose a hierarchical organisation of parafoveal processing, wherein low-level orthographic processing, facilitated by the visual word form area (VWFA), precedes higher-level semantic processing, supported by the left inferior frontal gyrus (LIFG).

## Discussion

The present study aims to understand how low-level word information, such as orthography, and high-level word information, such as semantics, are extracted before a word is fixated during natural reading. By applying representational similarity analysis to co-registered MEG and eye-tracking data, we found neuronal evidence that, ~68 ms after fixation onset on a word (Fig. 3a), orthographic processing of the parafoveal word is initiated in the visual word form area (VWFA) (Fig. 4b). Subsequently, semantic processing of the parafoveal word is initiated at ~137 ms (Fig. 3b), supported by the left inferior frontal gyrus (LIFG) (Fig. 4d). This hierarchical organisation allows for efficient pre-processing of different levels of parafoveal information, facilitating faster reading (Fig. 3c, d).

We provide direct neuronal evidence for parafoveal processing occurring at both orthographic and semantic levels, demonstrating that such parafoveal pre-processing is indeed deep. Although previous eye-tracking and ERP/FRP studies have provided evidence for orthographic and sometimes semantic parafoveal processing, they have largely approached this question indirectly. Eye-tracking studies[13–18,24,26,54,55] have inferred the existence of parafoveal processing by showing that exposure to a word's features in the parafovea leads to shorter fixation durations when the word is later fixated, i.e., already in the foveal region. Electrophysiological studies[29–40] have used a contrast design to compare event-/fixation-related potentials (ERPs/FRPs) for different types of parafoveal words, such as orthographic regular versus irregular or semantic congruent versus incongruent

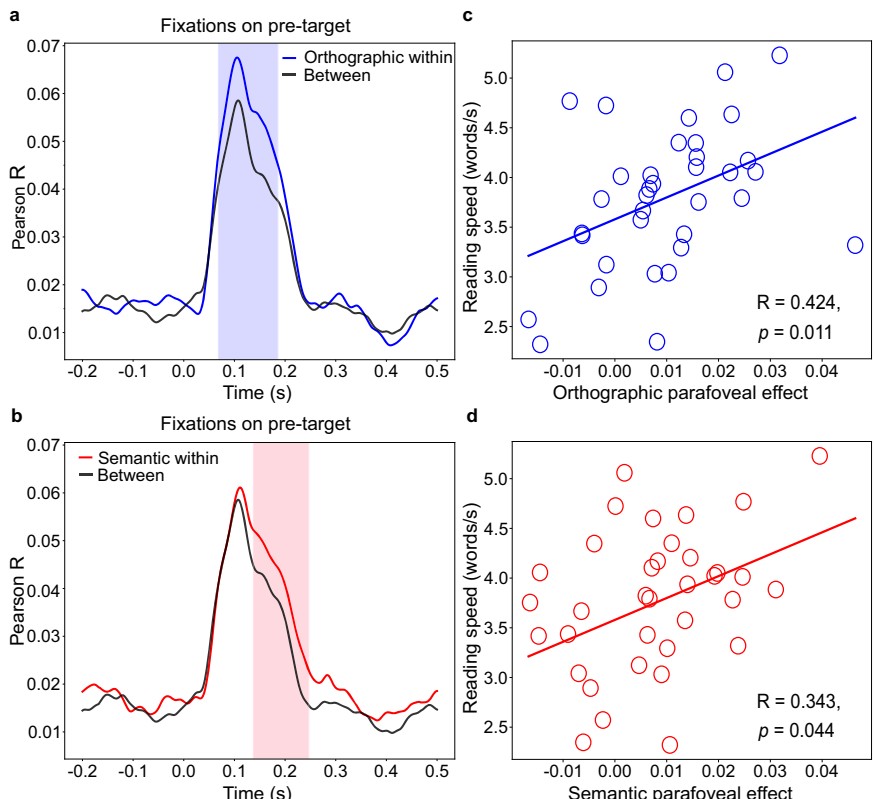

**Fig. 3 | Hierarchical parafoveal processing and its relationship with reading speed. a** Neuronal evidence for fast orthographic parafoveal processing. The time series of representational similarity (Pearson $R$-values) for orthographic within pairs (blue line) and between pairs (black line). The orthographic within pairs showed significantly higher representational similarity values than the between pairs during the 68–186 ms interval (indicated by the light blue shading) after fixation onset on pre-target words ($p < 0.001$; two-sided cluster permutation test). **b** Neuronal evidence for fast semantic parafoveal processing. The time series of representational similarity (Pearson $R$-values) for semantic within pairs (red line) and between pairs (black line). In the 137–247 ms interval after the fixation onset on pre-target words (indicated by the light red shading), the semantic within-pair target words showed significantly higher representational similarity values than the between pairs ($p < 0.001$; two-sided cluster permutation test). **c** Relationship

between the orthographic parafoveal effects and individual reading speed. Orthographic parafoveal effects were quantified by the mean difference in representational similarity values ($\Delta R$) between orthographic within pairs and between pairs. This was done within the time interval, revealing significant differences in the RSA analysis (68–186 ms after fixation onset on the pre-target words). The reading speed of each participant was quantified as the number of words read per second. Each dot represents one participant. The Spearman correlation revealed a positive correlation between the neuronal orthographic parafoveal effect and reading speed ($R = 0.42$, $p = 0.011$). **d** A Spearman correlation demonstrated that the neuronal semantic parafoveal effect positively correlated with individual reading speed ($R = 0.34$, $p = 0.044$). Source data are provided as a Source Data file.

words. These differences in ERPs/FRPs were used to infer the extraction of orthographic or semantic information from parafoveal words. Here, we employed a multivariate approach to directly analyse the distributed brain activity associated with parafoveal information. We thus provide neural evidence that both orthographic and semantic information can be extracted from the parafoveal word, supporting the notion of parallel processing across multiple words during natural reading[56]. Our approach was inspired by other studies that used EEG/ MEG to decode semantic representations associated with predictions in sentences presented word-by-word using RSA[48–51]. The RSA methodology seems to overcome some limitations of other multivariate techniques based on the classification of semantic word content[57].

It is important to note that we used the same pre-target word (e.g., "clever") in each sentence sextet (Fig. 1b), ensuring that differences in representational similarity between orthographic/semantic within and between pairs during pre-target fixation intervals are attributed to parafoveal processing of the target word, rather than foveal processing of the pre-target word. Moreover, the observed parafoveal effects were not influenced by the predictability of parafoveal words, as all target words had low cloze probability values. The semantic similarity of the context preceding the target words was also controlled (for details, see Behavioural pre-test in *Methods*) to mitigate potential contributions

from contextual semantic overlap. While these steps decrease the influence of extraneous factors, they cannot eliminate them entirely. One could also consider using the same sentence frames, so the linguistic context would be fully controlled. However, repeating identical sentence frames would introduce other problems. For instance, participants would recognise the repeated sentence frames and start predicting the upcoming words or become less attentive, resulting in a faster reading speed and increased skipping rate. Additionally, it is challenging to embed six target words into the same sentence without introducing semantic anomalies or unnatural phrasing. Therefore, using the same pre-targets within a sextet and presenting them far apart in the experiment was a design we considered optimal. It should be mentioned that a related design has been used in previous EEG and MEG studies using RSA to investigate prediction during reading, in which different sentence frames were also used[48–51]. It should also be noted that the target words used in our study are relatively short (mostly between 4 and 6 letters), the parafoveal effect observed in the current study may be reduced when parafoveal words are relatively longer, which deserves further investigation in future studies. Lastly, our design intentionally minimised the lexical variability of pre-target and target words to optimise the sensitivity of the RSA, but this approach also somewhat limits the insight into how various word-

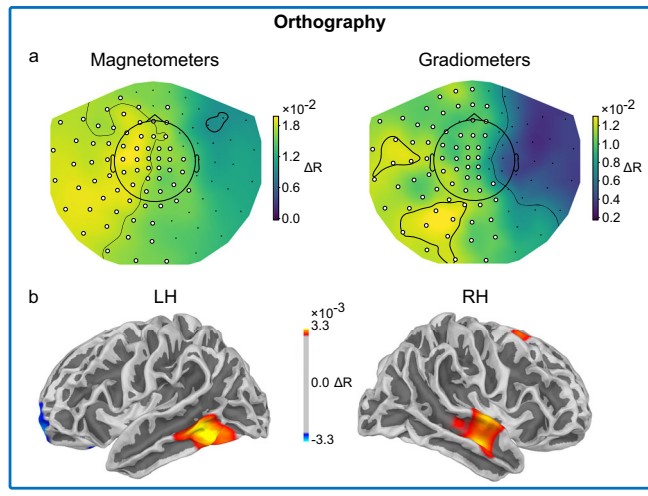

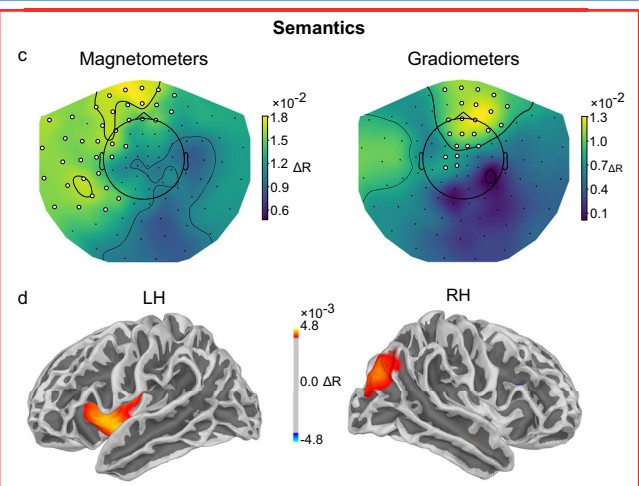

**Fig. 4 | Neuronal sources of orthographic and semantic parafoveal processing.**
**a** Mean sensor-level topographies across participants for the difference between representational similarity of orthographic within pairs and between pairs, in the 68–186 ms interval after fixation onset on the pre-target words, for magnetometers (left) and gradiometers (right), respectively. Sensors that showed significant differences in the correlation values were marked by white dots. **b** Source-level representational similarity analysis (RSA) results for the left and right hemispheres. The colour corresponds to the difference in representational similarity between orthographic within pairs and between pairs. The threshold was set at the 96th, 97.5th, and 99.95th percentiles of the data distribution, with sources exhibiting larger differences shown in orange and smaller differences in blue, colour coded based on their percentile. The location of the left cluster is consistent with the visual word-form area. **c** Topographic maps of the semantic parafoveal effect derived from a searchlight RSA. The figures on the left and right show the contribution of magnetometers and gradiometers to the semantic parafoveal effect. **d** Source-level RSA results for the left and right hemispheres. Clusters showing the maximal difference are in the left inferior frontal gyrus and around the right posterior parietal cortex. Source data are provided as a Source Data file.

characterised the N400-type ERPs/FRPs in response to parafoveal words that were semantically incongruent versus congruent with the sentence context[33–40]. How does the timing of the N400 (emerging at 250–300 ms) relate to the -137 ms onset of semantic parafoveal processing found in our study? As we typically saccade every 200–300 ms during natural reading, the parafoveal N400 emerges when the parafoveal word is often already fixated. The ERP/FRP paradigms may not be optimally sensitive to detecting early semantic effects, as they rely on stimulus-locked, averaged neural responses that are well suited to isolating robust components such as the N400. However, earlier neural computations that do not produce clear univariate ERP deflections may be missed. Indeed, previous studies have demonstrated that multivariate decoding can reveal condition-related differences at earlier latencies than those identified by univariate ERP analyses[59,60]. It is also worth noting that N400-type studies typically capture higher-level processes, such as the integration of the parafoveal target word into the sentence context, whereas our approach isolates the neuronal activity associated with word-level semantic access prior to integration into the broader context. That is, the semantic relationships between target words (e.g., writer–author) are context independent, which distinguishes our paradigm from classical N400 paradigms. The early timing of parafoveal orthographic and semantic processing observed in our study might also be partly explained by the naturalistic reading paradigm we adopted. In MEG studies where participants process isolated words[61], word processing of a target word *n* occurs after fixating on it. However, in our study, words in a sentence are simultaneously presented, allowing one or more words to be read prior to fixation. That is to say, processing of the target word may have been initiated at the *n*-2 word position (or even earlier), thereby facilitating faster access at word position *n*-1, a possibility that is not available in isolated word designs. This would be an important question to explore in future research.

Our results demonstrated that the different levels of parafoveal processing are staggered: low-level orthographic information is available first (-68 ms) (Fig. 3a), and higher-level semantic information is available relatively later (-137 ms) (Fig. 3b). Additionally, these processes overlap temporally: semantic processing begins while orthographic processing is reflected in the data until ~186 ms. This suggests that semantic processing may start once sufficient orthographic information (e.g., partial letter sequences) is available, while further orthographic detail is processed. Another possibility is that the overlap might reflect an intermediate stage of orthographic-to-phonological conversion (e.g., grapheme-phoneme mapping), which could co-occur with early semantic activation. The observed temporal overlap between orthographic and semantic processing supports a partially parallel model of word recognition rather than a strictly serial one. To the best of our knowledge, the current study is the first to compare the neuronal time course of parafoveal orthographic and semantic information extraction of the same word at the representational level during natural reading, revealing hierarchical parafoveal processing. It is worth noting that the same between pair (e.g., "writer" and "police") was used as a baseline for both orthographic and semantic within pair, which allowed us to directly compare the time course of orthographic and semantic parafoveal processing.

The neural sources underlying orthographic and semantic parafoveal processing were found to follow a hierarchical organisation. By applying a searchlight approach, we found RSA patterns in the left ventral occipitotemporal cortex (lvOT) associated with the processing of parafoveal orthographic information. The lvOT has been labelled the visual word-form area (VWFA)[53] as it plays a significant role in orthographic processing. Our findings extend prior insights into the lvOT by demonstrating that it is engaged not only in foveal but also in parafoveal orthographic processing. Our study also provided evidence that parafoveal semantic processing is supported by the left inferior

specific factors shape parafoveal processing in natural reading. Future work could extend our approach by incorporating a broader and systematically varied set of materials—modulating factors such as word length, frequency, and predictability—to establish the key factors and boundary conditions for parafoveal processing.

The timing of orthographic parafoveal processing (-68 ms) aligns well with the timing of visual information reaching the visual (-50 ms) and the temporal cortices (-70 ms)[58]. However, there is a debate as to when semantic information is derived. Previous work investigating the neuronal substrate of semantic parafoveal processing has

frontal gyrus (LIFG)—a region classically associated with various aspects of semantic processing, including the retrieval of lexical-semantic knowledge[62], semantic decision[63-65], semantic integration/unification[66,67] and semantic short-term memory[68,69]. The LIFG has also been reported as a key region for the representation of semantic similarity[70,71] and the abstractness dimension of word meaning[72]. However, other brain areas known to be involved in semantic processing, e.g., the left inferior temporal gyrus and the anterior temporal lobe, did not show evidence of representing semantic similarity in our study. This may be explained by MEG being less sensitive to regions (e.g., the ATL) further away from the sensors. Therefore, the absence of evidence of other regions known to be involved in semantic processing should not be over-interpreted.

We have thus far elucidated the time course and spatial localisation of orthographic and semantic representations extracted from parafoveal words. But what is the behavioural significance of these neural representations of parafoveal information? Our correlation results indicate that the ability to extract orthographic and semantic information from parafoveal words is positively correlated with individual reading speed. This supports the notion that parafoveal processing is necessary for fluent reading[2-4]. Our findings challenge strategies aimed at improving reading by reducing visual crowding, such as those employed for typical readers and individuals with dyslexia. For instance, rapid serial visual presentation (RSVP) methods, which display words in isolation, may hinder reading performance by limiting the ability to read and integrate several words per fixation.

In summary, we applied RSA to co-registered MEG and eye-tracking data to uncover the neuronal mechanisms associated with orthographic and semantic parafoveal processing during natural reading. We found that orthographic parafoveal processing emerges already at -68 ms after the fixation onset on the pre-target word, followed by semantic parafoveal processing at -137 ms. We further identified the VWFA and LIFG as the neural sources of orthographic and semantic parafoveal processing, respectively. This parafoveal processing was associated with individual reading speed, with stronger parafoveal effects observed in faster readers. Our results provide evidence for fast hierarchical parafoveal processing supported by the language network.

## Methods

### Participants
We recruited 39 native English speakers (24 self-reported as female), aged 21 ± 2.3 (mean ± SD). No sex or gender analysis was carried out, as these factors were not the focus of the current study. All participants were right-handed, had normal or corrected-to-normal eyesight, and had no known neurological or reading disorders (e.g., dyslexia). Four participants were excluded from the data analysis due to excessive head movement, poor eye tracking, or too many bad sensors during the recordings, leaving a total of 35 participants (22 females) for analysis. As we were embarking on an underexplored approach, a formal power analysis could not be conducted, but prior to the data acquisition, we estimated the number of participants based on prior studies using the pairwise RSA analysis to EEG and MEG data[48-51]. Participants provided written informed consent and were compensated £15 per hour for their participation. The study was approved by the University of Birmingham Ethics Committee (under the approved Programme ERN_18-0226P).

### Stimuli
The stimuli consisted of 360 plausible one-line sentences, each embedded with an unpredictable target word (the plausibility of the sentences and the unpredictability of the target words were confirmed by behavioural pre-tests, details provided below). The target words were always preceded and followed by at least 3 words within the sentences. The length of target words ranged from 3-8 letters (M = 5.1, SD = 1.2), with most words (92.6%) being 4-6 letters long. We constructed sentences in 120 triplets. Within each sentence triplet, the target word could be a critical word (e.g., "writer"), its orthographic neighbour (e.g., "waiter") differing by only one or two letters (116 instances with a one-letter difference, 4 with a two-letter difference), or its semantic neighbour (e.g., "author") with a highly similar meaning. Each sentence triplet was paired with another triplet that had a similar structure of target words (e.g., "police/policy/guards") (see Fig. 1b). This pairing established orthographic within-pair (e.g., "writer-waiter") and semantic within-pair relationships (e.g., "writer-author"), while also providing unrelated between-pair controls (e.g., "writer-police"/"writer-policy"/"writer-guards", randomly chosen in analysis). Within each sentence sextet (a group of six sentences formed by pairing two triplets), the target words had identical lengths. To ensure a similar level of processing of pre-target words when processing different target words in the parafovea, the pre-target words were identical (e.g., "clever") within a sentence sextet. Please note that only sentences in a sextet shared the same pre-target words. These 6 sentences were presented separately during the experiment, with an average of 58 other sentences (minimum 35) appearing between any two of them. Additionally, to control for order effects, the 360 sentences were divided into two halves (180 each). For half of the participants, the second half was presented first, and vice versa.

### Behavioural pre-test
**Predictability test of the target words.** We carried out a cloze norming task to assess the predictability of the target word given the prior context in each sentence. The task involved 25 participants (18 self-reported as female), all of whom were native English speakers, aged 24.0 ± 4.3 years (mean ± SD), with no reading disorder. None of the participants participated in the MEG experiment. The data were collected via the online survey platform Qualtrics (https://www.qualtrics.com). Participants were presented with the sentence frames up to the target word and were asked to predict the next word in the sentence. See below an example sentence frame for the target word "waiter":

They gave the clever __________

If more than 25% of participants predicted the target word, then this target word was considered predictable. If more than 65% of participants predicted the same word, though not the target word used in the experiment, the sentence was considered highly constrained. One target word was judged to be highly predictable, and one sentence was highly constrained, so the two sentences were modified and retested with 24 different participants (14 males). In the final version of sentences, the average predictability of the target words was 0.9 ± 3.0% (mean ± SD), indicating that the target words were unpredictable; and the average predictability of the most frequently predicted non-target words was 20.3 ± 10.4 % (mean ± SD), suggesting that the sentence contexts were not highly constrained.

**Plausibility test.** A separate group of participants (24 in total, 14 self-reported females, aged 22.2 ± 1.8, mean ± SD, one participant's data were excluded due to random responses) participated in the plausibility test. The data were collected via Qualtrics (https://www.qualtrics.com/) using a 7-point rating scale. The test included 360 experimental sentences, along with 200 filler sentences—100 implausible and 100 anomalous—to occupy the full range of the plausibility scale. Participants were instructed to read each sentence and rate its plausibility: (1) if the sentence did not make any sense, (4) if it was unlikely but still possible, and (7) if it was fully acceptable. Before beginning the task, participants were shown example sentences (not included in the actual test) with corresponding ratings and were encouraged to use the full

scale. In the example below, the first sentence was from our experimental material, while the second and third sentences were implausible and anomalous, respectively.

| | Implausible Plausible |
|---|---|
| They gave the clever waiter a tip for his good service. | 1 2 3 4 5 6 7 |
| The man used a kettle to cook porridge yesterday evening. | 1 2 3 4 5 6 7 |
| Jeremy quenched his thirst with a glass of programme. | 1 2 3 4 5 6 7 |
| ...... | 1 2 3 4 5 6 7 |

The average plausibility rating for the experimental sentences was $6.0 \pm 0.5$ (mean ± SD), which was significantly higher than implausible filler sentences ($3.2 \pm 0.8$ mean ± SD; two-tailed paired $t$-test: $t_{(22)} = 25.94$, $p < 0.001$, Cohen's $d = 5.41$) and anomalous filler sentences ($1.8 \pm 0.6$ mean ± SD; two-tailed paired $t$-test: $t_{(22)} = 17.39$, $p < 0.001$, Cohen's $d = 3.63$).

**Semantic relatedness.** In natural language, words with similar meanings tend to appear in similar contexts. To ensure that the representational similarity was driven solely by the similarity of the target words and not by their preceding contexts, we evaluated the context similarity by computing the latent semantic analysis (LSA) values (http://wordvec.colorado.edu/) between the contexts of the target words. For example, to assess the context similarity of the target words "police" and "guards", we computed the LSA value between the contexts "The man was found by the clever" and "The manager assigned two clever".

The results showed no difference in the semantic similarity of the contexts between orthographic within pairs and unrelated between pairs (two-sided independent samples $t$-test: $t_{(238)} = 0.74$, $p = 0.462$, Cohen's $d = 0.10$). Similarly, there was no difference in the semantic similarity of the contexts for semantic within pairs and unrelated between pairs (two-sided independent-samples $t$-test: $t_{(238)} = 0.95$, $p = 0.343$, Cohen's $d = 0.12$). Thus, the representational similarity for orthographically or semantically similar parafoveal words could not be explained by the semantic similarity in the context.

## Experimental procedure

The experiment took place in a dimly lit, magnetically shielded room, where participants were seated under the MEG gantry. The gantry was set at a 60° upright angle and covered the participants' heads entirely. A projection screen, positioned 145 cm from the participants' eyes, was used to display 360 one-line sentences, which were programmed using Psychophysics Toolbox-3[73] in MATLAB R2019b (Mathworks Inc., USA). The sentences were shown in bold, black text (RGB: [0, 0, 0]), using size-32 Courier New font with equal spacing, on a neutral grey background (RGB: [128, 128, 128]). Each letter and space occupied 0.316 degrees of visual angle, and the sentences typically spanned between 12.64 and 25.60 visual degrees in width. Participants were instructed to silently read the sentences at their own pace while minimising head and body movement. Their eye movements and brain activity were recorded simultaneously using an eye tracker and MEG.

Each trial started with a fixation cross at the centre of a middle-grey screen, lasting for 1.2–1.6 s. Then a black square (0.4° wide) was presented at the vertical centre of the screen, 1.3 degrees of visual angle from the left edge. Sentence presentation was triggered when participants fixated on this square for at least 0.2 s, with the sentence starting from the location of the 'starting square'. During sentence presentation, a grey "ending square" (RGB: [64, 64, 64], 0.4° wide) was displayed below the centre of the screen, 2.6 degrees of visual angle from the left edge. After reading the sentence, participants fixated on the ending square for 0.1 s to terminate the sentence presentation, followed by a 0.5-s blank middle-grey screen. To ensure careful reading, 25% of all sentences were followed by a simple yes-or-no comprehension question (Fig. 1a). Participants scored 90% or better in response to the questions (mean accuracy = 96.3%, SD = 2.6%). The experiment consisted of ten blocks, with each block containing 36 sentences and taking approximately 5 minutes to read. Following each block, participants were given a rest period of at least 30 s. In total, the experiment took about 1 h.

## Data acquisition

**MEG.** MEG data were obtained using a 306-sensor TRIUX Elekta Neuromag system (Elekta, Finland), consisting of 204 orthogonal planar gradiometers and 102 magnetometers. Three bony fiducial points (nasion, left and right preauricular points) were digitised utilising the Polhemus Fastrack electromagnetic digitiser system before the MEG recording. The digitisation process was then extended to include four head-position indicator (HPI) coils, placed on the left and right mastoid bones and on the forehead, ensuring a minimum distance of 3 cm between the two forehead coils. Additionally, over 250 additional points, evenly distributed across the entire scalp, were digitised to assist in aligning the MEG head model with individual structural MRI images. The MEG data were online filtered between 0.1 and 330 Hz with anti-aliasing filters and sampled at 1000 Hz.

**Eye movements.** During the MEG session, we used the EyeLink 1000 Plus eye tracker (long-range mount, SR Research Ltd, Canada) to collect participants' eye movement data. The eye tracker was placed on a wooden table between the participants and the projection screen. The distance from the centre of the participant's eyes to the camera of the eye-tracker was 90 cm. The eye tracker's centre was aligned with the middle of the projection screen, and its top edge reached the bottom of the screen. With a 1000 Hz sampling rate, the eye-tracker continuously recorded both horizontal and vertical eye positions, as well as pupil size from the left eye of each participant. To ensure accurate and reliable tracking of eye movements, the experiment started with a nine-point calibration and validation test, which aimed to achieve an accuracy level where the permitted tracking error was below 1 visual degree in both horizontal and vertical dimensions. To maintain accuracy throughout the session, we conducted a one-point drift check every three trials and before the start of each block. Additionally, any failure to trigger sentence presentation through gaze led to an immediate one-point drift-check. If the drift-check error was bigger than 2 degrees, a nine-point calibration and validation were performed.

We extracted fixation events from the EyeLink output file. To ensure that our neural measures reflect parafoveal processing upon the first encounter with the pre-target word, we selected only first fixations on the pre-target word during the first pass of reading. If a participant initially skipped the word and later returned to fixate on it, those trials were excluded from the analysis. These steps eliminated potential contamination from later regressive fixations and potential confounds from delayed or secondary processing unrelated to parafoveal processing. Fixations that were too long (>1000 ms) or too short (<80 ms) were also excluded from the analysis. Eye movement measures, including first fixation duration, landing position, and refixation rate on pre-target words, are reported in the Supplementary Materials (See Supplementary Fig. 3).

We also recorded Electrooculography (EOG) data by placing one pair of electrodes approximately 2 cm away from the outer canthus of each eye for horizontal EOG recordings, and another pair above and below the right eye in line with the pupil for vertical EOG recordings.

**MRI.** Following MEG data acquisition, participants were scheduled for a separate visit to have an MRI scan. The T1-weighted structural MRI image was acquired with a 3-Tesla Siemens Magnetom Prisma scanner (TR = 2000 ms, TE = 2.01 ms, TI = 880 ms, flip angle = 8°, FOV = 256 × 256 × 208 mm, isotropic voxel size = 1 mm). For the 7 participants who dropped out of the MRI scan, we utilised the standard FreeSurfer (version 6) average subject named "fsaverage" to do source modelling.

## MEG data analyses

**Pre-processing.** MEG data were analysed using MNE Python[74] (version 1.3.0) and following the FLUX pipeline[75] (https://www.neuosc.com/flux). First, we identified sensors with excessive artefacts using a semi-automatic detection algorithm (on average, about 6 faulty sensors per participant). Signal-space separation (SSS) and Maxwell filtering[76] were then applied to reduce artefacts from environmental sources and sensor noise. Faulty sensors were repaired via SSS, ensuring that all 306 MEG sensors were ultimately used for each participant. The data were down-sampled to 200 Hz and bandpass filtered at 1–40 Hz prior to performing independent component analysis (ICA). The fast ICA algorithm[77] was then applied to decompose the data into 30 independent components. Components containing ocular (eye blinks and eye movements), and heartbeat artefacts were identified by manually viewing the time course and topographies of the ICA components. Additionally, components containing ocular artefacts were confirmed by detecting those most correlated with the EOG signals using Pearson correlation. These identified components were subsequently removed from the original raw data (which was not downsampled or filtered). Next, we segmented the MEG data into epochs that contained a time window of −200 to 500 ms relative to first fixation onsets on the pre-target word. We then applied a 30 Hz low-pass filter to eliminate high-frequency noise from the epoched data.

**Representational similarity time course analysis.** To combine gradiometers and magnetometers in the RSA analysis, we first normalised the MEG signals using z-scores for each sensor. For each trial, at each time point (from −200 to 500 ms relative to the fixation onset on the pre-target word, e.g., "clever"), we extracted the MEG signals across all 306 sensors, creating a $1 \times 306$ vector. This vector captures the spatial pattern of neural activity at a specific time point $t$ (see Fig. 2). To also capture the temporal activation pattern, we applied a sliding time window approach[78,79], incorporating data from $t − 32$ ms to $t + 32$ ms. This resulted in a $306 \times 65$ matrix (sensors × time points) for each time point. The matrix for each time point was then flattened into a $1 \times 19,890$ vector representing the neural activity pattern surrounding time $t$. We calculated the representational similarity between pairs of trials by computing Pearson's correlation between their corresponding vectors. This was done separately for three conditions: orthographic within pairs, semantic within pairs, and between pairs. The number of pairs for each condition was similar: $85.6 \pm 14$, $84.5 \pm 13.7$, and $85.0 \pm 13.9$ (mean ± SD, across participants) for orthographic within pairs, semantic within pairs, and between pairs. We then averaged the correlation values across all pairs within each condition at $t$ to obtain $R_{orth}(t)$, $R_{sema}(t)$, and $R_{between}(t)$. This process was repeated for each time point within the −200–500 ms interval relative to the first fixation onset on the pre-target words, resulting in 3 time series of pairwise correlations: $R_{orth}$, $R_{sema}$, and $R_{between}$ for each participant. To visualise the data, we averaged these similarity values across all participants ($N = 35$) at each time point. This yielded a grand-average spatial similarity time series for each condition (See Fig. 3a, b).

**Sensor-level searchlight representational similarity analysis.** To investigate which sensors contributed to the observed orthographic and semantic parafoveal effects, we applied a searchlight approach[46] to conduct the representational similarity analysis (RSA) in the sensor space (Fig. 4a, b). This analysis was conducted separately for magnetometers and gradiometers. For each magnetometer, a searchlight patch was defined by including 20 neighbouring magnetometers, while for each gradiometer, the searchlight patch included 40 neighbouring gradiometers, as gradiometers are arranged in pairs. The analysis was conducted within specific time windows identified from the representational similarity time course analysis (see Fig. 3a, b): 68–186 ms for the orthographic parafoveal effect, and 137–247 ms for the semantic parafoveal effect, both aligned with the onset of the first fixation on the pre-target word. For each sensor, we extracted MEG data from each trial within the relevant time window, producing an $M \times N$ matrix where $M$ represents the number of sensors in the searchlight patch (20 for magnetometers, 40 for gradiometers), and $N$ represents the number of time points (119 for orthography and 111 for semantics, given the 1000 Hz sampling rate). This matrix was flattened into a $1 \times (M \times N)$ vector, representing the neural activity pattern within the searchlight patch during the respective time window. We then computed representational similarity for orthographic within pairs by calculating Pearson correlations between corresponding pairwise vectors. This produced a set of $R$ values, which were averaged to obtain the representational similarity for orthographic within pairs ($R_{orth}$). A similar process was used for between pairs to obtain the average $R$ values for between pairs ($R_{between}$). The difference between $R_{orth}$ and $R_{between}$ for each sensor was used to quantify its contribution to the orthographic parafoveal effect. This analysis was repeated for all sensors, generating a topographical map of each sensor's contribution to the observed orthographic parafoveal effect (Fig. 4a). Similarly, we calculated the Pearson correlation for the semantic within pairs ($R_{sema}$) and contrasted it with the $R_{between}$ within the 137–247 ms time window for each sensor to generate the topographical map of each sensor's contribution to the observed semantic parafoveal effect (Fig. 4b).

**Source reconstruction.** First, FreeSurfer[80] was used to automatically reconstruct the cortical surfaces from participants' anatomical MRI images. Co-registration between the MRI and MEG coordinate frames was done using the three fiducial landmarks obtained from the head shape digitisation process. For 7 participants who did not have individual anatomical images, a standard template (i.e., FreeSurfer average brain "fsaverage") was warped to fit the participant's head shape, estimated from the digitised points. A surface-based source space, consisting of 20,484 vertices evenly distributed across the cortical surface (10,242 per hemisphere), was generated for each participant. The head conductivity model was built using individual structural MRIs and was modelled as a single-layer boundary element model (BEM). The forward solution was then computed using the transformation matrix between the MEG "head" and MRI coordinate frames, the source space and the BEM model.

A noise-covariance matrix was estimated using the −1000–0 ms intervals relative to the fixation cross before sentence presentation. The forward solution, implemented as a lead-field matrix, along with the noise-covariance matrix enabled the estimation of the inverse solution for source localisation. We computed the inverse solution minimum norm estimates using the dynamic statistical parametric mapping (dSPM) approach[81] for single epochs, which contained a time window of −200 ms to 500 ms relative to first fixation onsets on pre-target words. The epochs were downsampled to 100 Hz to reduce computation times. To perform group-level analysis, individual source estimates were morphed onto a common source space (i.e., "fsaverage") before performing source-level RSA.

As for the searchlight RSA in the source space, we used the same approach as in the sensor space, except that each searchlight patch comprised the 2000 closest vertices for a given vertex.

**Statistical analysis.** To assess whether and when significant differences in the within-pair (for both orthographic and semantic) and between-pair correlations emerged while controlling for multiple

comparisons over time, we employed a cluster-based permutation test[82]. This approach creates a surrogate distribution by taking contiguous time-points of significance into account by randomly permuting the data. We first computed differences between within-pair and between-pair correlations, focusing on specific time intervals of interest: 60–250 ms after fixating on the pre-target words, resulting in two contrast arrays: one for orthography ($D_{orth}$) and another for semantics ($D_{sema}$). To identify time points when these contrasts significantly deviated from the null hypothesis (i.e., no effect), we performed one-sample $t$-tests across participants at each time point. Clusters of adjacent time points with significant $t$-statistics ($p < 0.05$, two-sided) were identified. Next, we performed permutation tests by randomly flipping the sign of differences (i.e., $D_{orth}$ and $D_{sema}$). This process was repeated 5000 times to construct a null distribution of the maximum cluster statistic (i.e., the sum of $t$-values within a cluster) under each permutation. The observed cluster-level statistics were then compared against this null distribution. Clusters falling within the highest or lowest 2.5% of the null distribution indicated significant differences in the representational similarity between conditions (i.e., $R_{orth}$ and $R_{between}$ or $R_{sema}$ and $R_{between}$).

This statistical analysis for the sensor-level searchlight RSA was performed in a similar manner to the RSA time course analysis, with the key difference being that it focused on the spatial patterns of the data rather than time points.

### Reporting summary

Further information on research design is available in the Nature Portfolio Reporting Summary linked to this article.

## Data availability

The following data in the current study can be found on figshare (https://doi.org/10.6084/m9.figshare.27189843)[83]: the raw MEG data, pre-processed epoched data, the raw EyeLink files, the Psychotoolbox data, and the head models after the co-registration of T1 images with the MEG data. The raw T1-weighted MRI images are not shared to protect participant privacy. Source data are provided with this paper.

## Code availability

The experiment presentation scripts (Psychtoolbox) and analysis codes for the paper are available on OSF[84].

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

## Acknowledgements

This work was supported by the following grant to O.J.: Wellcome Trust Discovery Award (grant number 227420) and by the NIHR Oxford Health Biomedical Research Centre (NIHR203316). Y.P. was supported by the Leverhulme Early Career Fellowship (ECF-294-2023-626). L.W. acknowledges support from the China Scholarship Council (CSC) for providing a PhD scholarship. The views expressed are those of the author(s) and not necessarily those of the funders. The funders had no role in the preparation of the manuscript or the decision to publish.

## Author contributions

L.W., S.F., Y.P., and O.J. devised and designed the study, L.W. made the sentences with assistance from S.F., L.W. collected and analysed the data with assistance from Y.P., O.J. and S.F. All authors contributed to writing the paper.

## Competing interests

The authors declare no competing interests.
