## [Transparent Peer Review file · Nature Communications]

Fast hierarchical processing of orthographic and semantic parafoveal information during natural reading

Corresponding Author: Professor Ole Jensen

Version 0:

Reviewer comments:

Reviewer #1

(Remarks to the Author)

Review of NCOMMS-24-64929 - Fast hierarchical processing of orthographic and semantic parafoveal information during natural reading

Simon P. Liversedge

This paper reports a single experiment investigating parafoveal processing during natural reading. Co-registration methodology was used whereby MEG and eye movement recordings were taken as participants read sentences normally. The experimentation in this paper is high quality and the theoretical questions are interesting, important and relevant to researchers in the broad field of reading. I do feel that the results deserve to be published, however, there are a number of theoretical overstatements in the paper and I have a number of methodological concerns. To reiterate, though, I feel that these data should be published. I detail the thoughts that occurred to me as I read the paper.

P2, line 15. The sentence starting "Readers extract..." at the beginning of the Abstract is contentious, and arguable. Almost all researchers investigating reading and eye movements agree that orthographic information is extracted from parafoveal words. However, whether semantic properties of parafoveal words are extracted has been, and is, a topic of tremendous debate. I estimate that at least half the researchers working in this area would fundamentally disagree with this statement. I encourage the authors to look at the work of Brothers et al. and Zang et al. to see very strong experimental tests of semantic parafoveal processing. Brothers et al. conclude categorically that no such effects exist.

T. Brothers, L.J. Hoversten, M.J. Traxler. Looking back on reading ahead: No evidence for lexical parafoveal-on-foveal effects *Journal of Memory and Language*, 96 (2017), pp. 9-22

Zang, C., Zhang, Z., Zhang, M., Degno, F. & Liversedge, S.P. (2023). Examining semantic parafoveal-on-foveal effects using a Stroop boundary paradigm. *Journal of Memory and Language*, 128, 2023, 104387, ISSN 0749-596X. <https://doi.org/10.1016/j.jml.2022.104387>.

p3, Line 42. This sentence is inaccurate. Hundreds of studies investigating parafoveal processing have orthogonally manipulated two or more variables, that is, more than one level of word information.

P5, Line 78. The description of the RIFT method is inaccurate. The technique is described as involving the presentation of subliminally flickering words and it is claimed that the participant does not perceive the flicker. The authors were recently good enough to invite myself and colleagues to visit their laboratory and experience the RIFT method in person. When I was tested, I could very clearly identify the words that were flickering. In my view, this flicker could cause participants to allocate attention to upcoming words in text in a way that they would not if those words were not flickering. And in our discussions after the laboratory demonstration, I think it is fair to say that colleagues at Birmingham acknowledged that this point represented a reasonable concern. I don't want to make a big deal about this, because the RIFT method is very interesting and valuable and the results using this method demonstrate, at least in principle, that readers may be sensitive to aspects of words in the parafovea. Whether similar effects might occur under more natural reading circumstances (e.g., using a boundary paradigm with MEG co-registration) is an interesting question. The key point I suggest the authors take on is to be accurate in their description of the method. If they simply toned down claims about the subliminal nature of the flicker and don't state that participants are not aware of the flicker, then there is no problem.

p6, Line 109. Here the two theoretical questions under examination are provided. The second question is framed in terms of "temporal hierarchy", that is whether the orthographic and semantic characteristics of a parafoveal word become available simultaneously (i.e., in parallel), or sequentially, orthographic first followed by semantic. This issue is a very well-studied aspect of parafoveal processing that has been examined over decades. The issue concerns the time course with which different types of information about a parafoveal word become available (most often indexed via eye movement effects). In describing these issues in terms of a processing hierarchy, the authors are engaging in "rebranding". There is no need to use this new terminology here - instead, the discussion sits in the context of a well-established literature on the time course of processing.

Also, how could a reader gain access to the semantic characteristics of a word without first establishing a representation of its orthographic or phonological form? A word's identity (or at least a set of candidates) must be established in order that semantic characteristics associated with the word(s) might be accessed. As far as I can see, in natural reading, this has to happen via processing of orthographic or phonological form (i.e., such processing must necessarily occur prior to access of semantic characteristics). If the authors disagree with my characterisation here, then I suggest the onus sits with them to explain their alternative perspective. If they agree with my suggestion here, perhaps they might change their theoretical question?

p7. Here I have a number of experimental concerns. First, at a general level, no eye movement data are presented in the paper. This is very surprising to me given that the paper focuses on the neural processes that occur as participants make a series of fixations across words (i.e., read). The lack of consideration of the eye movement results suggests that those data are something of an irrelevance. However, this is not the case at all. When we read, we make saccades and fixations. Each fixation is a discrete episode of visual delivery to the brain. This means that the precise location of a fixation (with respect to the linguistic stimulus) and its duration is determinant in respect of what might be available to be processed visually and cognitively during that particular period (or episode). For example, the position of a fixation within a word will significantly influence the degree to which the upcoming word is processed effectively in the parafovea. Additionally, the length of the fixated word will influence this too. Whether the word preceding the parafoveal word is fixated once or refixated very substantially effects parafoveal processing. The length of the upcoming word will affect processing. The launch site distance is also known to affect the degree to which a parafoveal word is processed. Because the eye movement data were not considered in any detail in the analyses, it is unclear whether any of these considerations were taken into account. Furthermore, we cannot know whether the basic effects were modulated by these factors. My intention in making this point is not to be obstructive (to reiterate, I think that this work should be published). Instead, I think that there are a wealth of theoretical issues that the authors have not considered, but which could be assessed via the current data set. There are a multitude of theoretically interesting questions here. Bottom line, I think that the neural data need to be considered much more carefully in relation to the eye movement data. Again, the precise pattern of fixations on word are a determining factor in relation to the nature of processing that might occur.

A second point concerns the stimuli that were constructed. The authors make an amount of how carefully the stimuli were constructed (pre-target word was the same across conditions; LSA conducted to ensure similar meaning of prior context; cloze predictability norming for the target words). However, the stimuli are non-optimal for this kind of experiment. As can clearly be seen from Figure 1, the sentential frames were completely different across conditions. This means that effects observed at the pre-target and target words, potentially, could occur due to differences in sentential context rather than due to the experimental manipulations. Of course, the authors realise that this issue is something of a problem and this is why the pre-target word was kept the same, and the LSA and cloze analyses conducted. These steps, whilst admirable, do not prevent the possibility that differences in the sentential frames could have contributed to the effects that are reported. By the way, carrying out LSA is not a generally accepted method for ensuring semantic comparability across linguistic stimuli. Moreover, from my perspective, it is hard to see why the authors did not simply create stronger stimuli for their experiment. For example, if we take the 6 target words (writer, waiter, author, police, policy, guards) that the authors used in their example stimulus, they can easily be embedded in the same sentence frame (e.g., "They recognised the clever ***** almost immediately."). If the stimuli had been constructed in this way, and the pre-screen procedures that have been adopted here had also been carried out, the experiment would have been much stronger and it would be clear that the effects could not be caused by differences in sentential content prior to the target word.

By the way, it would be helpful if the full stimulus set could be provided on the OSF website - I could not find the stimuli when I looked on-line.

Power analyses. It is increasingly the case that the sample size for participants and stimuli should be determined through power analyses. I may have missed it, but I did not see such analyses. The authors might wish to provide such analyses.

RSA Analyses. I have to admit that I am no expert on these analyses, but one thing that I wanted to check was whether the analyses take into account that 700 successive comparisons will be made across the time window and this approach affords the possibility of alpha inflation. I'm sure that the authors have this covered, but I felt I should raise it just in case.

P8. The results show differences in activation for the orthographic relative to the semantic conditions across two time windows with different time courses (earlier for orthographic, slightly later for semantic). As I've already noted, this relative difference in the time course of effects is exactly what would be expected. However, the critical issue concerns whether these effects are observed when the pre-target word is under fixation. As we know, fixations on the pre-target region will vary in their durations - some will be as short as 120ms or so, most will be around 200ms (as the authors note) and others will persist for longer. The temporal range of the semantic effects is 137-247ms. Unless I am mistaken, this means that for a significant proportion of the fixations that contributed to this effect, the eyes were fixating the target word for at least some of

the period of time that has been analysed. That is to say, the effects occurred when the target word was in foveal, not parafoveal, vision. I think that this is a very important issue. And without seeing more of the eye movement data, and analyses that consider the two data sets in relation to each other, it is impossible to know whether this could represent a serious problem. I think that the analyses should be reconducted to examine only those fixations when the eyes were actually fixating the pre-target region for the entire period during which the effect occurred (in other words, shorter pre-target fixations where the eyes moved to the target during the temporal window under scrutiny should be removed from the analyses). Also, if my reasoning is correct here, then the statement on P9, line 170 (intersaccadic interval) is not strictly accurate.

P9. Another point here is that the neural effects are examined entirely in relation to the pre-target word. And, of course, the effects are taken to reflect characteristics of the target word that affect processing one word earlier. However, shouldn't the effects persist beyond the pre-target word? I mean, if anything when the target word is fixated, we should see similar, but much more concrete, effects of the type observed at the pre-target word since the target word is now fully clearly visible in foveal vision. To be clear, shouldn't there be some persistence of the effects from the pre-target to the target word?

p10, line 183. There is some loose use of language here. There is discussion of some individuals extracting more orthographic and semantic information than others. Actually, the results showed larger orthographic and semantic effects for some than others - this does not necessarily imply extraction of more such information.

I think that the statement in the first paragraph at the top of P12 goes beyond the data. I do not accept that this is a model - as a "model" it has no computational formality and it is actually just a linguistic description.

There was a degree of repetition in the Discussion section and I felt it was a little disjointed (e.g., P15). Perhaps the authors could re-read and edit a little to make it flow better. Unsurprisingly, I disagreed with the statement at the bottom of P13 over onto the top of P14 - I'm not as confident as the authors regarding the possibility that the effects could not arise from extraneous factors. And in the following paragraph, again, there is no need to talk about the effects in relation to a "temporal hierarchy". Instead, just say that orthographic characteristics come available with a faster time course than semantic characteristics (note again that this is not a particularly remarkable finding).

The statements regarding interventions on P16 were (to me) unnecessary and unrealistic. Are the authors really suggesting that remediation techniques for less able readers should involve training in parafoveal processing efficiency?? This sounds like a 1970's throwback. I strongly suggest this be removed.

p17. I note that the target words varied quite a lot in terms of length. This will also determine (to some degree) how effectively they were parafoveally processed. This underlines the importance of considering the neural data in relation to the eye movement data in more detail. Were the effects similar for long and short words - I think that the authors must predict that the parafoveal effects would be reduced for long relative to short words, right?

P20. The characterisation of the plausibility assessment is inaccurate here. Three types of sentence were adopted: grammatically legal plausible sentences (the experimental sentences); grammatically legal but implausible sentences; grammatically illegal sentences. The latter category cannot be evaluated in respect of their plausibility since they do not have a grammatically legal form. In fact, I cannot understand how the participants could have been required to form a judgment as to the plausibility of word strings with this form - can the authors explain this? I note that these issues were considered in detail in the paper by Rayner et al. (2004).

Rayner, K., Warren, T., Juhasz, B., & Liversedge, S.P., (2004). The Effect of Plausibility on Eye Movements in Reading. *Journal of Experimental Psychology Learning Memory and Cognition* 30(6):1290-301.

Minor points

p3, Line 36. "Change "to-be-focused" to "to-be-fixated". This is not an issue of accommodation.

p4, Lines 55 & 56. Delete "the" from "the preview benefit". I suggest this change be applied throughout.

p4, Line 70. This sentence should be rewritten. The effect does not exert any influence over saccadic planning. The effect reflects influences on processing.

p6, Line 108. Change "fixated upon" to "fixated". Again, I suggest this change be applied throughout.

P18, Line 344. The analyses should be re-run without the 4 stimuli that differ by two letters, not just one. This is a standard procedure for eye movement experiments.

p18, Line 351. Change "ensure a same" to "ensure a similar level of".

P19, Line 376. Even though the predictability levels were quite low, they still sit at 20%. This is actually quite a high value - I'm surprised by this. I suggest that this receives some comment in the text.

A number of the references in the Reference section are not formatted correctly and do not provide full reference information (e.g., 1&2 and 19&20 inconsistently list the same journal; 6, 8, 11, 25, 41, 44 and 45 are incomplete).

Reviewer #2

(Remarks to the Author)

The authors report MEG evidence to suggest that, during reading, orthographic and semantic information is extracted from parafoveal words around ~70ms and ~140ms, respectively.

Summed up like that in a single sentence, this may not sound like much, but it actually is. This is an excellent way of

exploiting MEG: the combination of the preview boundary paradigm with representational similarity analysis (RSA) is clever, and thanks to the high spatio-temporal resolution of MEG produces a very clear picture of the timecourse of parafoveal information extraction. In my view the RSA approach provides an important new angle on a longstanding debate, and therefore I deem this study worthy of publication in Nature Communications after minor revisions. I've added some comments below.

Joshua Snell

p.3 line 42: It is claimed that most studies have focused on a single level of word information. That is true. However, in Snell, Meade, et al. (2019, *Neuropsychologia*) we showed correlates of parafoveal processing both at sub-lexical and lexical levels. Perhaps the authors mean to say that most studies have manipulated only one word dimension (e.g., orthographic flanker overlap in aforementioned study); but that single manipulation may nonetheless inform multiple levels of processing.

p.4 line 58: The authors claim that preview benefit effects do not reveal the timecourse of parafoveal information extraction, but some preview benefit studies have in fact addressed this; see e.g. the 'fast-priming' paradigm of Hohenstein, Laubrock & Kliegl, 2010, in *JEP:LMC*. The authors may note that the method of Hohenstein et al. had a shortcoming though: they replaced previews at some point during the fixation on the pre-target. These sudden changes in the visual field likely capture attention and thus produce a distorted image of how much parafoveal processing there typically is. Long story short, the timecourse of parafoveal processing is not a new topic; but novel methods are certainly welcome.

p.5 line 74: "Earlier neural mechanisms reflecting the onset of parafoveal semantic processing may exist but have yet to be identified." In Snell et al. (2023, *Cortex*), we report FRP effects caused by parafoveal grammatical violations as early as 100ms after fixating the preceding word. The authors talk specifically about semantics, but I'd say syntax and semantics go hand in hand: to notice the grammatical violation, readers had to extract some meaning from the parafovea.

At several points in the Intro the authors highlight the fact that they track orthographic *and* semantic information extraction *from the same word*. Maybe that's not the novelty that needs to be emphasized. Firstly, many preview benefit studies have similarly used various preview conditions to assess various levels of processing in a single experiment. Secondly, it would be equally fine to have several separate studies reporting on different levels of processing; together they provide a complete picture. I think you're OK by just emphasizing the novelty of the RSA approach.

Methods: The authors could justify the use of the boundary paradigm a bit more. For the RSA, you just need contrasts between neighbors and unrelated words - and one might argue that you don't need the boundary paradigm at all for this. Just inspect neural activity during the fixation on 'clever' in

- the clever waiter
- the clever writer
- the clever author
- the clever scouts

see where and when contrasts [waiter-writer] and [waiter-author] are different from [waiter-scouts] and you're done?

I anticipate the justification would be that you want all conditions to be perfectly equal once the eyes leave 'clever', but most of the magic happens before that moment anyway, right?

Results: Although you had significant differences, it is quite striking that, overall, correlations were always really low (Pearson $R < 0.07$). A few words on this would be good.

Reviewer #3

(Remarks to the Author)

This is a review of the paper titled "Fast hierarchical processing of orthographic and semantic parafoveal information during natural reading", submitted for publication in Nature Communications by Lijuan Wang et al..

This is a signed review. I'm Davide Crepaldi (SISSA, Trieste and the University of Pavia), and wrote the following notes with substantial help from my postdoc Giulio Severijnen, in the context of the (brilliant, in my view) Early Career Researcher co-reviewing initiative of the journal.

Overall, this seems like a nice, solid paper, which brings good theoretical insight and is methodologically sound. The experiment is ingenious, and does provide important information about word processing during natural reading in a way that previous studies couldn't. The paper is also very nicely written – clear, compelling, concise. All in all, neither me nor Giulio see big obstacles to publication in NatComm.

However, we do have a number of theoretical and methodological points that we'd like the authors to take up in a revision – some of which we see as quite major.

Re: the theoretical framing of the study, I'm not super happy with the authors' characterization of the eye tracking, invisible-boundary findings. They give it for granted that semantic parafoveal effects are well established in that literature, and present them as essentially on par with orthographic and phonological effects (around line 52). I'm not necessarily on top of the latest development in this field (so apologies if I'm missing something crucial here), but my understanding is that while there's wide agreement that orthography is processed in the parafovea, eye tracking experts are still strongly debating the existence of semantic effects.

To my eyes, also the FRP data characterization is a bit questionable. The authors take the fact that FRP semantic effects appear relatively late — later than what would be necessary for that information to influence the upcoming saccades — as reason to believe that there's an earlier component that has gone unnoticed thus far (around line 70). This logic only stands if one assumes that parafoveal words' semantics must be part of the processing leading to saccade decisions. The possibility is not considered that this might simply not be true — and therefore, the FRP semantic effect is all there is to observe and is, so to speak, "appropriately late".

Now, there is nothing objectively wrong in the authors' approach, and of course the present results do show (and convincingly so) that there was indeed an earlier component. However, the storytelling seems to give much a priori justification for the results that are eventually found — more than the pre-existing literature would justify, I would contend. To be clear — this does not change my conviction that the results are interesting and solid. I don't believe that data are epistemologically stronger when they are more strongly expected *ex ante*. Still, I would personally appreciate a more balanced layout of the landscape for this study.

Partly related to the previous point, the semantic effects that are found here are really quite early — probably earlier than most of the MEG literature on semantic processing of printed words. To illustrate with an example that I know well :), Vignali et al. (2023) found that the earliest semantic component in an MEG study on isolated printed word arises around 300ms post stimulus onset. In the same study, word frequency — which tracks lexical identification and is therefore widely thought to precede meaning computation — emerged no earlier than 300ms post. Of course, we can see reasons why semantic processing might have emerged earlier in the present work (e.g., sentence embedding allows prediction), but also reasons why it might have emerged later (e.g., foveal processing of isolated words, as in Vignali et al., must be more perceptually powerful than parafoveal processing). Again, this has implications for the rationale of the present experiment (*ex ante*), and for the discussion of the present results (*ex post*). The authors discuss quite in depth the connection between the present data and FRP-derived N400 (around line 270); less or no emphasis is given to the integration of the current findings with MEG-derived timelines for visual word processing more generally.

There's a correlation between the magnitude of the parafoveal effect(s) and reading speed. This is interesting and makes perfect sense. However, correlation isn't causation, and we see plausibility in both causal directions — quicker readers make more use of parafoveal information to support their speed, or readers with better grasp of parafoveal information can then read more quickly. The authors use a wording (e.g., "predict") that might imply a causal interpretation, and might seem to privilege one causal direction over the other (e.g., lines 323-324). However, this isn't entirely clear; maybe the authors can reformulate more explicitly and/or elaborate more?

A final theoretical point concerns the content of the orthographic and semantic computations that the authors have uncovered, and their overlap. This was mostly triggered by the authors' explanations for the timing of the effects. They show that the orthographic effect takes place between 68-186 ms and the semantic between 137 - 247 ms. In the discussion (around line 266), they conclude that this timing aligns well with visual information reaching the visual (50 ms) and temporal (70 ms) cortices. While the lower boundary makes sense, we thought it remains a bit unclear what happens *until* 186 ms. What type of orthographic processing is still happening at 186 ms? The authors present in several passages what might seem as a quite serial view of printed word processing. So, we're wondering what they would make of the overlap between the orthographic and the semantic windows.

A similar kind of argument can be made for the semantic effect. The authors state that semantic processing is well aligned with the estimated timing constraints, allowing to affect the decision to skip the target (p.14, line 277). But we don't really know that participants actually do this — effectively use this semantic information to inform saccade decisions.

This is probably where the authors' natural reading paradigm shows its limitations; while it is certainly very ecological and does provide a new look into these theoretical questions, it's difficult to go deeper in terms of how (and whether, in fact) the information that the brain encodes is then used cognitively. Maybe the authors can find room to discuss this point?

On more methodological grounds, we have the following questions/suggestions.

We wonder whether the authors have addressed the possibility that on some trials, their participants might have skipped ahead within the core of the sentences right when the sentences appeared, to then return at the beginning. I have seen quite a lot of this in my own natural reading studies. Although we wouldn't know what kind of processing these early run-ahead fixations might trigger, this might of course mess up the timing a bit (e.g., participants might have had some early orthographic processing?).

Again on the methods, while most of the details are thoroughly discussed, we wonder about the specifics of the stimuli and the experimental procedure. For example, the authors do provide one example of a sextet (so two triplets that have the same pre-target word), but we seem not to be able to find the entire stimulus list (also not on OSF). It would be nice to have it somewhere.

Related to this, we wonder if any effects could be explained by the specificity of the triplet pairings. Each triplet was matched with another triplet, after which the data were averaged. This is appropriate of course. However, was there any substantial item variability? Did some triplets show larger effects than others? Are there triplets that don't show the effect? It would be good to get a feel for how much the triplet pairings contributed to variance in the results (over and above the general guarantee given by the statistics).

Within every triplet, the same control sentence is being used to compare orthographic and semantic effects (e.g., O: writer - waiter, C: writer-police; S: writer-author, C: writer-police). Perhaps it would be good to check if there aren't any effects due to choosing one specific control sentence. For example, the authors might run the same analyses, but with a different control sentence (e.g., writer-policy)?

Finally, regarding the experimental procedure more generally, we couldn't find how the experimental lists were randomized. The authors have 360 sentences, each of which is shown once. This is divided into 10 blocks of 36 sentences. Was this arrangement fully random? Did participants see sentences of the same triplet in the same block? And what about the order of the sentences? Did participants for example always see first the target sentence (writer) and then the orthographic (waiter) and semantic one (author)? Or was this randomized, too? The reason we ask is that perhaps there could be some spill-over effects from one sentence to the other. Maybe the neural response (in terms of RSA, but also in terms of the timing of the effects) on waiter/author is different when participants first saw "writer", compared to when they did not?

Reviewer #4

(Remarks to the Author)

Version 1:

Reviewer comments:

Reviewer #1

(Remarks to the Author)

I would like to thank the authors for considering my comments and for taking the time to revise their manuscript to deal with the points I raised. I think that they have done a very good job in their revisions and responses. I think that the revised manuscript is much stronger and I would like to see it published. Again, I am grateful to the authors for responding to my points.

Simon P. Liversedge

(Remarks on code availability)

Reviewer #2

(Remarks to the Author)

I was already quite enthusiastic about the original manuscript. The authors have taken great care to improve the clarity of the paper, and I think they did a good job responding to the points raised by myself as well as the points raised by the other reviewers. I'm happy to recommend publication of the paper in its current form.

Joshua Snell

(Remarks on code availability)

Reviewer #3

(Remarks to the Author)

We thank the authors for their responsiveness to our comments. They addressed all our issues satisfactorily and we now fully endorse the publication of this paper.

We'd only have two remaining minor points that we'd like to ask some clarification on.

On p.16, the authors state that "The E/FRP paradigms may not be sensitive enough to detect an early semantic effect, as they rely on phase-locked, averaged neural responses that may miss transient or subtle dynamics". However, it is unclear to us why this is the case. Are the authors suggesting that ERP paradigms are not fit to detect early effects? We understand that the analyses used in ERP studies can make it difficult to detect subtle, oscillatory effects, but how is this related to the timing of ERPs?

The authors then continue by stating that "It is also worth noting that N400-type studies typically capture higher-level processes, such as the integration of the parafoveal target word into sentence context, whereas our approach isolates the neuronal activity associated with parafoveal semantic information at the single-word level". Here again it is unclear to us why this is the case. First, we're doubting why ERP studies should necessarily capture higher-level processes. Second, how then is the present study different from previous studies? Since the present study still involves integration into a sentence context.

We don't necessarily need to see a revised version of the manuscript, but we think providing more substantial argumentation for these points would improve the manuscript further. Thank you again for this interesting work!

(I co-reviewed the revised manuscript again with my post-doc Giulio Severijnen.)

(Remarks on code availability)

Reviewer #4

(Remarks to the Author)

(Remarks on code availability)

REVIEWER COMMENTS

Reviewer #1 (Remarks to the Author):

Review of NCOMMS-24-64929 - Fast hierarchical processing of orthographic and semantic parafoveal information during natural reading

Simon P. Livversedge

This paper reports a single experiment investigating parafoveal processing during natural reading. Co-registration methodology was used whereby MEG and eye movement recordings were taken as participants read sentences normally. The experimentation in this paper is high quality and the theoretical questions are interesting, important and relevant to researchers in the broad field of reading. I do feel that the results deserve to be published, however, there are a number of theoretical overstatements in the paper and I have a number of methodological concerns. To reiterate, though, I feel that these data should be published. I detail the thoughts that occurred to me as I read the paper.

We sincerely appreciate the reviewer's constructive comments as well as positive feedback on the quality of our experiment and the theoretical significance of our study. We have carefully considered the reviewer's comments and made corresponding revisions to further improve the manuscript. Below, we provide detailed responses to each of the reviewer's points.

P2, line 15. The sentence starting "Readers extract..." at the beginning of the Abstract is contentious, and arguable. Almost all researchers investigating reading and eye movements agree that orthographic information is extracted from parafoveal words. However, whether semantic properties of parafoveal words are extracted has been, and is, a topic of tremendous debate. I estimate that at least half the researchers working in this area would fundamentally disagree with this statement. I encourage the authors to look at the work of Brothers et al. and Zang et al. to see very strong experimental tests of semantic parafoveal processing. Brothers et al. conclude categorically that no such effects exist.

T. Brothers, L.J. Hoversten, M.J. Traxler. Looking back on reading ahead: No evidence for lexical parafoveal-on-foveal effects *Journal of Memory and Language*, 96 (2017), pp.

Zang, C., Zhang, Z., Zhang, M., Degno, F. & Liversedge, S.P. (2023). Examining semantic parafoveal-on-foveal effects using a Stroop boundary paradigm. *Journal of Memory and Language*, 128, 2023,104387, ISSN 0749-596X. <https://doi.org/10.1016/j.jml.2022.104387>.

Thanks – indeed we should have phrased this more carefully. The opening sentence of the Abstract has been revised to reflect this ongoing debate (p.2, line 15 in the revised manuscript):

“In reading, information from parafoveal words is extracted before direct fixation; however, it is debated whether this processing is restricted to orthographic features or encompasses semantics. Moreover, the neuronal mechanisms supporting parafoveal processing remain poorly understood. We co-registered MEG and eye-tracking data in a natural reading paradigm to uncover the timing and brain regions involved in parafoveal processing.”

Furthermore, we have added the studies on lexical and semantic Parafoveal-on-foveal effects to the Introduction:

“Other eye movement studies have investigated parafoveal processing by measuring which characteristics of parafoveal words influence the processing of the currently fixated word, i.e., parafoveal-on-foveal (PoF) effects. While orthographic PoF effects are well established (for a review see¹), the results from eye movement studies have largely not provided evidence in favour of lexical and semantic PoF effects (Brothers et al., 2017; Zang et al., 2023). Taken together, eye-tracking evidence remains inconclusive as to whether semantic features can be extracted from the parafovea”

p3, Line 42. This sentence is inaccurate. Hundreds of studies investigating parafoveal processing have orthogonally manipulated two or more variables, that is, more than one level of word information.

We thank the reviewer for pointing out this inaccuracy, we have removed the sentence.

P5, Line 78. The description of the RIFT method is inaccurate. The technique is described as involving the presentation of subliminally flickering words and it is claimed that the participant does not perceive the flicker. The authors were recently good enough to invite myself and colleagues to visit their laboratory and experience the RIFT method

in person. When I was tested, I could very clearly identify the words that were flickering. In my view, this flicker could cause participants to allocate attention to upcoming words in text in a way that they would not if those words were not flickering. And in our discussions after the laboratory demonstration, I think it is fair to say that colleagues at Birmingham acknowledged that this point represented a reasonable concern. I don't want to make a big deal about this, because the RIFT method is very interesting and valuable and the results using this method demonstrate, at least in principle, that readers may be sensitive to aspects of words in the parafovea. Whether similar effects might occur under more natural reading circumstances (e.g., using a boundary paradigm with MEG co-registration) is an interesting question. The key point I suggest the authors take on is to be accurate in their description of the method. If they simply toned down claims about the subliminal nature of the flicker and don't state that participants are not aware of the flicker, then there is no problem.

We thank the reviewer for raising this concern about the subliminal nature of RIFT. Since the current study does not employ frequency tagging, we have revised our description of RIFT to tone down claims about the subliminal nature of the flicker (p.5, line 86 in the revised manuscript):

“This technique involves flickering the location of the parafoveal word at a high frequency, such as 60 Hz. Tagging responses of the visual flicker can be detected in the brain and used to measure the degree of attention allocated to the parafoveal word.”

p6, Line 109. Here the two theoretical questions under examination are provided. The second question is framed in terms of "temporal hierarchy", that is whether the orthographic and semantic characteristics of a parafoveal word become available simultaneously (i.e., in parallel), or sequentially, orthographic first followed by semantic. This issue is a very well-studied aspect of parafoveal processing that has been examined over decades. The issue concerns the time course with which different types of information about a parafoveal word become available (most often indexed via eye movement effects). In describing these issues in terms of a processing hierarchy, the authors are engaging in "rebranding". There is no need to use this new terminology here - instead, the discussion sits in the context of a well-established literature on the time course of processing.

We thank the reviewer for the valuable feedback regarding terminology. In posing the theoretical questions, we originally used "serially or in parallel" (p.6, line 112 in the old

manuscript) to frame the temporal organisation of parafoveal processing. However, we did use the term "temporal hierarchy" in the Discussion. We have revised the manuscript to avoid this phrasing, now stating (p.17, line 329 in the revised manuscript):

“Our results demonstrated that the different stages of parafoveal processing are staggered: low-level orthographic information is available first (~68 ms) (Fig. 3a) and higher-level semantic information is available relatively later (~137 ms) (Fig. 3b).”

Also, how could a reader gain access to the semantic characteristics of a word without first establishing a representation of its orthographic or phonological form? A word's identity (or at least a set of candidates) must be established in order that semantic characteristics associated with the word(s) might be accessed. As far as I can see, in natural reading, this has to happen via processing of orthographic or phonological form (i.e., such processing must necessarily occur prior to access of semantic characteristics). If the authors disagree with my characterisation here, then I suggest the onus sits with them to explain their alternative perspective. If they agree with my suggestion here, perhaps they might change their theoretical question?

We agree with the fundamental premise that semantic processing follows orthographic or phonological processing, which is well-supported by existing literature about word processing. Our study does not challenge this foundational principle but instead investigates the temporal and spatial organisation of these processes during parafoveal processing in natural reading. We've revised the research question part in our manuscript (p.7, line 114 in the revised manuscript):

“Our study aims to address two core questions: (1) Can we identify specific representational activity associated with orthographic and semantic information of a parafoveal word before it is fixated? (2) If so, what are the neuronal time course and brain areas associated with orthographic and semantic parafoveal processing?”

p7. Here I have a number of experimental concerns. First, at a general level, no eye movement data are presented in the paper. This is very surprising to me given that the paper focuses on the neural processes that occur as participants make a series of fixations across words (i.e., read). The lack of consideration of the eye movement results suggests that those data are something of an irrelevance. However, this is not the case at all. When we read, we make saccades and fixations. Each fixation is a discrete episode of visual delivery to the brain. This means that the precise location of a fixation (with respect to the linguistic stimulus) and its duration is

determinant in respect of what might be available to be processed visually and cognitively during that particular period (or episode). For example, the position of a fixation within a word will significantly influence the degree to which the upcoming word is processed effectively in the parafovea. Additionally, the length of the fixated word will influence this too. Whether the word preceding the parafoveal word is fixated once or refixated very substantially effects parafoveal processing. The length of the upcoming word will affect processing. The launch site distance is also known to affect the degree to which a parafoveal word is processed. Because the eye movement data were not considered in any detail in the analyses, it is unclear whether any of these considerations were taken into account. Furthermore, we cannot know whether the basic effects were modulated by these factors. My intention in making this point is not to be obstructive (to reiterate, I think that this work should be published). Instead, I think that there are a wealth of theoretical issues that the authors have not considered, but which could be assessed via the current data set. There are a multitude of theoretically interesting questions here. Bottom line, I think that the neural data need to be considered much more carefully in relation to the eye movement data. Again, the precise pattern of fixations on word are a determining factor in relation to the nature of processing that might occur.

We thank the reviewer for raising this important point regarding the presentation of eye movement data in our study. In response, we have now included supplementary analyses of the eye-tracking data, including first fixation durations, landing position, and refixation rate for pre-target words. Please note that we cannot present the eye movement data by the experimental conditions, i.e. orthographic-within, semantic-within, and between pair conditions, because a pair of fixations rather than a single fixation. However, the overall data show typical reading behaviours (see Supplementary Figure 3). These results are reported in the revised manuscript (see p.27, line 541) and presented in Supplementary Figure 3.

“Eye movement measures, including first fixation duration, landing position, and refixation rate on pre-target words, are reported in the Supplementary Materials (See Supplementary Figure 3).”

Supplementary Figure 3 | Eye movement characteristics for fixations on pre-target words.

(a) The first fixation durations on the pre-target words. The points represent the average duration for the first fixations for individual participants. The red dashed line indicates the grand average (N = 35). The box represents the interquartile range (IQR), with the central line marking the median. Outliers are shown as grey circles. (b) The average landing position on first fixations on pre-target words (normalised from 0 to 1 according to physical word length). (c) Refixation rate (%) of pre-target words.

We did take eye movement patterns into account in the RSA analysis; for instance, we only included first fixations during the first pass of reading, as outlined in the revised manuscript (p.27, line 534) (p.27, line 534):

“To ensure that our neural measures reflect parafoveal processing upon the first encounter with the pre-target word, we selected only first fixations on the pre-target word during the first pass of reading. If a participant initially skipped the pre-target word and later returned to fixate it, those trials were excluded. These steps eliminate potential contamination from later regressive fixations and potential confounds from delayed or secondary processing unrelated to parafoveal processing. Fixations that were too long (>1000 ms) or too short (<80 ms) were also excluded from the analysis.”

Regarding the influence of word length of foveal/parafoveal words on parafoveal processing, we have added the following text to the Discussion (p.15, line 294 in the revised manuscript).

“It should also be noted that the target words used in our study are relatively short (mostly between 4 to 6 letters), the parafoveal effect observed in the current study may be reduced when parafoveal words are relatively longer, which deserves further investigation in future studies. Lastly, our design intentionally minimised the lexical

variability of pre-target and target words to optimise the sensitivity of the RSA, but this approach also somewhat limits the insight into how various word-specific factors shape parafoveal processing in natural reading. Future work could extend our approach by incorporating a broader and systematically varied set of materials—modulating factors such as word length, frequency, and predictability—to establish the key factors and boundary conditions for parafoveal processing.”

Lastly, we appreciate the reviewer’s point about how eye movement characteristics influence parafoveal processing. However, our pairwise MEG-RSA is based on pairs of fixation events, while eye movement measures are fixation-specific. Each pair yields one neural similarity measure (from MEG), but two sets of eye movement data (one per fixation), complicating attempts to directly examine whether eye movement measures modulate the observed parafoveal effects.

A second point concerns the stimuli that were constructed. The authors make an amount of how carefully the stimuli were constructed (pre-target word was the same across conditions; LSA conducted to ensure similar meaning of prior context; cloze predictability norming for the target words). However, the stimuli are non-optimal for this kind of experiment. As can clearly be seen from Figure 1, the sentential frames were completely different across conditions. This means that effects observed at the pre-target and target words, potentially, could occur due to differences in sentential context rather than due to the experimental manipulations. Of course, the authors realise that this issue is something of a problem and this is why the pre-target word was kept the same, and the LSA and cloze analyses conducted. These steps, whilst admirable, do not prevent the possibility that differences in the sentential frames could have contributed to the effects that are reported. By the way, carrying out LSA is not a generally accepted method for ensuring semantic comparability across linguistic stimuli. Moreover, from my perspective, it is hard to see why the authors did not simply create stronger stimuli for their experiment. For example, if we take the 6 target words (writer, waiter, author, police, policy, guards) that the authors used in their example stimulus, they can easily be embedded in the same sentence frame (e.g., "They recognised the clever ***** almost immediately."). If the stimuli had been constructed in this way, and the pre-screen procedures that have been adopted here had also been carried out, the experiment would have been much stronger and it would be clear that the effects could not be caused by differences in sentential

content prior to the target word.

By the way, it would be helpful if the full stimulus set could be provided on the OSF website - I could not find the stimuli when I looked on-line.

We thank the Reviewer for this comment.

We have now uploaded the full stimulus list to OSF.

Indeed, the use of different sentence frames introduces potential variability.

Nevertheless, different sentence frames were an inevitable consequence of our experimental design as repeating identical sentence frames would introduce another set of problems. We have added a limitation section in the Discussion in the revised manuscript (see p.14, line 282).

“While these steps decrease the influence of extraneous factors, they cannot eliminate them entirely. One could also consider using the same sentence frames, so the linguistic context would be fully controlled. However, repeating identical sentence frames would introduce other problems. For instance, participants would recognise the repeated sentence frames and start predicting the upcoming words or becoming less attentive, resulting in a faster reading speed and increased skipping rate. Additionally, it is challenging to embed six target words into the same sentence without introducing semantic anomalies or unnatural phrasing. Therefore, using the same pre-targets within a sextet and presenting them far apart in the experiment was a design we considered optimal. It should be mentioned that a related design has been used in previous EEG and MEG studies using RSA to investigate prediction during reading in which different sentence frames were also used⁴⁸⁻⁵¹.”

Power analyses. It is increasingly the case that the sample size for participants and stimuli should be determined through power analyses. I may have missed it, but I did not see such analyses. The authors might wish to provide such analyses.

We appreciate the reviewer’s suggestion regarding power analysis. Due to the limited number of prior MEG studies using RSA, particularly in natural reading paradigms where trial loss occurs due to word skipping, well-established effect size estimates are not available. As a result, conducting a conventional power analysis was challenging. Instead, we referred to prior studies employing similar RSA methods⁴⁸⁻⁵¹ in controlled reading tasks to guide our sample size selection. We’ve added the following statement to our revised manuscript (p.20, line 394):

“As here we are embarking on a new approach, a formal power analysis could not be conducted, but prior to the data acquisition, we estimated the number of participants based on prior studies using the pairwise RSA analysis to EEG and MEG data⁴⁸⁻⁵¹”

RSA Analyses. I have to admit that I am no expert on these analyses, but one thing that I wanted to check was whether the analyses take into account that 700 successive comparisons will be made across the time window and this approach affords the possibility of alpha inflation. I'm sure that the authors have this covered, but I felt I should raise it just in case.

This is an important point we could have explained better. We confirm that multiple comparisons over time points were controlled using a cluster-based permutation test as explained in Maris & Oostenveld (2007), which has become a standard for human EEG and MEG data analysis. We have edited the Statistical analysis part in Methods to make it clearer (p.33, line 654 in the revised manuscript):

“To assess whether and when significant differences in the within-pair (for both orthographic and semantic) and between-pair correlations emerged while controlling for multiple comparisons over time, we employed a cluster-based permutation test (Maris & Oostenveld, 2007). This approach creates a surrogate distribution by taking contiguous time-points of significance into account by randomly permuting the data.”

P8. The results show differences in activation for the orthographic relative to the semantic conditions across two time windows with different time courses (earlier for orthographic, slightly later for semantic). As I've already noted, this relative difference in the time course of effects is exactly what would be expected. However, the critical issue concerns whether these effects are observed when the pre-target word is under fixation. As we know, fixations on the pre-target region will vary in their durations - some will be as short as 120ms or so, most will be around 200ms (as the authors note) and others will persist for longer. The temporal range of the semantic effects is 137-247ms. Unless I am mistaken, this means that for a significant proportion of the fixations that contributed to this effect, the eyes were fixating the target word for at least some of the period of time that has been analysed. That is to say, the effects occurred when the target word was in foveal, not parafoveal, vision. I think that this is a very important issue.

And without seeing more of the eye movement data, and analyses that consider the two data sets in relation to each other, it is impossible to know whether this could represent a serious problem. I think that the analyses should be reconducted to examine only those

fixations when the eyes were actually fixating the pre-target region for the entire period during which the effect occurred (in other words, shorter pre-target fixations where the eyes moved to the target during the temporal window under scrutiny should be removed from the analyses). Also, if my reasoning is correct here, then the statement on P9, line 170 (intersaccadic interval) is not strictly accurate.

This is an important point. To address the reviewer's concern, we conducted a control analysis excluding trials where the eyes moved to the target word within 247 ms of pre-target fixation onset. Please note that this analysis is quite strict because it also excludes trials that might contribute to the early parafoveal semantic effect. For instance, a pre-target fixation lasting 180 ms could support a semantic effect between 137–180 ms. With this strict trial exclusion criteria, we still observed a significant semantic parafoveal effect (141–195 ms, $p = .041$, cluster-based permutation test), and again the effect onset is ~140 ms, please see Supplementary Figure 1. This confirms that the observed semantic parafoveal effect is not the result of contamination from fast saccades resulting in early foveal processing of the target word. The results have been incorporated into the revised manuscript (see p.9, line 175) and are illustrated in Supplementary Figure 1.

“One potential concern is that the observed semantic effect (137–247 ms) may, in some cases, include early foveal processing of the target word, as short pre-target fixations could allow the eyes to move to the target during this window. To address this, we conducted a control analysis in which we excluded trials where fixation shifted to the target word within 247 ms, the semantic parafoveal effect remained significant ($p = .041$; cluster permutation test), again emerging at ~140 ms (see Supplementary Figure 1), confirming that the effect is not attributable to foveal processing of the target word.”

Supplementary Figure 1 | Controlled representational similarity analysis for first fixations on pre-target words.

The time series of representational similarity (Pearson R-values) for semantic within-pairs (red line) and between-pairs (grey line), with trials excluded where the eyes moved to the target word within 247 ms of pre-target fixation onset. The effect remained significant in the 141–195 ms interval ($p = .041$; cluster-based permutation test).

P9. Another point here is that the neural effects are examined entirely in relation to the pre-target word. And, of course, the effects are taken to reflect characteristics of the target word that affect processing one word earlier. However, shouldn't the effects persist beyond the pre-target word? I mean, if anything when the target word is fixated, we should see similar, but much more concrete, effects of the type observed at the pre-target word since the target word is now fully clearly visible in foveal vision. To be clear, shouldn't there be some persistence of the effects from the pre-target to the target word?

We appreciate the reviewer's insightful question about the persistence of neural effects beyond the pre-target fixation. We've conducted RSA analyses time-locked to the onset of fixations on the target and post-target words to address this question. We've added these analyses to the revised manuscript as below (p.10, line 187) and Supplementary materials:

"To understand the dynamics of word representations, we also conducted RSA analyses time-locked to the onset of fixations on the target and post-target words (see Supplementary Figure 2 for results)."

Supplementary Figure 2 | The RSA analysis for fixations on target and post-target

(a) Time series of RSA (Pearson's r values) aligned to fixation onset on target words. Representational similarity is shown for orthographic within-pairs (blue), semantic within-pairs (red), and between-pairs (black). First, the similarity for orthographic and semantic within-pairs exceeded the between-pairs at about 60–80 ms after fixation onset on the target word, while these effects did not reach statistical significance (orthographic: 60–89 ms, $p = 0.083$; semantic: 60–76 ms, $p = 0.189$; cluster-based permutation test). Later, in the 109 – 167 ms interval (indicated by light red shading), the semantic within-pairs produced significantly higher R values than the between pairs ($p = .023$; cluster-based permutation test). (b) Time series of RSA aligned to fixation onset on post-target words. No significant difference was observed between orthographic within-pairs and between-pairs (74–80 ms, $p = 0.29$; cluster-based permutation test). However, semantic within-pair similarity was significantly higher than between-pair similarity during the 60–133 ms interval (shaded in light red; $p = 0.008$; cluster-based permutation test).

"The dynamics of word representations

To examine whether the observed effects persisted during fixations on the target word, we conducted supplementary RSA analyses time-locked to the fixation onset on the target word. First, we observed that the similarity for orthographic and semantic within-pairs exceeded the between-pairs at about 60–80 ms after fixation onset on the target word, while these effects did not reach statistical significance (orthographic: 60–89 ms, $p = 0.083$; semantic: 60–76 ms, $p = 0.189$; cluster-based permutation test). Then, the orthographic effects were no longer observed as the orthographic within and between-pair similarities became comparable. This suggests that orthographic information extraction has already been completed during parafoveal processing and decayed thereafter, as it becomes irrelevant for ongoing sentence comprehension. Interestingly,

we observed an unexpected reversal effect at about 100 ms: semantically related target words (e.g., “writer” vs “author”) exhibited lower neural representational similarity than unrelated pairs, with a significant difference in the 109–167 ms interval ($p = .021$, cluster-based permutation test). A possible explanation could be related to the depth of processing. Parafoveal semantic processing is coarse and likely at gist-level. For example, the parafoveal words “writer” and “author” might be represented as “someone who produces written text” with similar neuronal representations. However, when these words are fixated, more detailed semantic processing is allowed, enabling finer differentiation between similar concepts. For example, a “writer” might refer to a person who has typed some text; while an “author” might refer to someone who has published written material. Consequently, during foveal processing, the neural “representational distance” between “writer” and “author” should be greater, even more so than that of dissimilar words, to ensure they are sufficiently differentiated from each other. Notably, this reversed pattern was specific to the target fixation: when analyses were time-locked to the subsequent fixation (i.e., after the target word), semantic similarity effects reappeared in the expected direction, with higher similarity for within-pairs than between-pairs (60–133 ms; $p = .011$, cluster-based permutation test). This dynamic change in word representations indicates that words are processed at different levels across several saccades, which may optimise the information extraction from words. However, the above explanation is tentative and needs further exploration.”

p10, line 183. There is some loose use of language here. There is discussion of some individuals extracting more orthographic and semantic information than others. Actually, the results showed larger orthographic and semantic effects for some than others - this does not necessarily imply extraction of more such information.

We agree that our original wording was not good, and we have revised the language (see p.11, line 200 in the revised manuscript).

“Our analysis revealed that both orthographic and semantic parafoveal effects significantly correlated with individual reading speed (orthography: Fig. 3c, $R = 0.42$, $p = .011$; semantics: Fig. 3d, $R = 0.34$, $p = .044$, Spearman correlation), suggesting that individuals with stronger orthographic and semantic parafoveal effects tended to be faster readers.”

I think that the statement in the first paragraph at the top of P12 goes beyond the data. I do not accept that this is a model - as a “model” it has no computational formality and it is actually just a linguistic description.

We acknowledge that this terminology could imply a formal computational model, and we have revised the text on page 12, line 239.

“We thus propose a hierarchical organisation of parafoveal processing, wherein low-level orthographic processing, facilitated by the Visual Word Form Area (VWFA), precedes higher-level semantic processing, supported by the Left Inferior Frontal Gyrus (LIFG).”

There was a degree of repetition in the Discussion section and I felt it was a little disjointed (e.g., P15). Perhaps the authors could re-read and edit a little to make it flow better. Unsurprisingly, I disagreed with the statement at the bottom of P13 over onto the top of P14 - I'm not as confident as the authors regarding the possibility that the effects could not arise from extraneous factors. And in the following paragraph, again, there is no need to talk about the effects in relation to a "temporal hierarchy". Instead, just say that orthographic characteristics come available with a faster time course than semantic characteristics (note again that this is not a particularly remarkable finding).

We thank the reviewer for his comments. We have carefully reviewed the Discussion section and made revisions to improve its flow and reduce repetition. As discussed in our response to the earlier point regarding stimulus construction, we have now added the use of different sentence frames as a potential limitation in the manuscript and have accordingly moderated our claims. For example, we have removed the sentence stating, *“Therefore, the present findings cannot be explained by facilitatory effects of predictability or semantic similarity in context.”* The revised text (p15, line 274) now reads:

“It is important to note that we used the same pre-target word (e.g., “clever”) in each sentence sextet (Fig. 1b), ensuring that differences in representational similarity between orthographic/semantic within- and between-pairs during pre-target fixation intervals are attributed to parafoveal processing of the target words, rather than foveal processing of the pre-target word. Moreover, the observed parafoveal effects were not influenced by the predictability of parafoveal words, as all target words had low cloze probability values. The semantic similarity of the context preceding the target words was also controlled (for details, see Behavioural pre-test in Methods) to mitigate potential contributions from contextual semantic overlap. While these steps decrease the influence of extraneous factors, they cannot eliminate them entirely. One could also consider using the same sentence frames so the linguistic context would be fully controlled. However, repeating identical sentence frames would introduce other problems. For instance, participants would recognise the repeated sentence frames and

start predicting the upcoming words or becoming less attentive, resulting in a faster reading speed and increased skipping rate. Additionally, it is challenging to embed six target words into the same sentence without introducing semantic anomalies or unnatural phrasing. Therefore, using the same pre-targets within a sextet and presenting them far apart in the experiment was a design we considered optimal. It should be mentioned that a related design has been used in previous EEG and MEG studies using RSA to investigate prediction during reading, in which different sentence frames were also used⁴⁸⁻⁵¹.

Additionally, we have avoided the terminology "temporal hierarchy" in the Discussion (see p.17, line 329 in the revised manuscript).

"Our results demonstrated that the different stages of parafoveal processing are staggered: low-level orthographic information is available first (~68 ms) (Fig. 3a) and higher-level semantic information is available relatively later (~137 ms) (Fig. 3b)."

The statements regarding interventions on P16 were (to me) unnecessary and unrealistic. Are the authors really suggesting that remediation techniques for less able readers should involve training in parafoveal processing efficiency?? This sounds like a 1970's throwback. I strongly suggest this be removed.

We have revised the text regarding interventions as below (p.19, line 372):

"Our findings challenge strategies aimed at improving reading by reducing visual crowding, such as those employed for typical readers and individuals with dyslexia. For instance, rapid serial visual presentation (RSVP) methods, which display words in isolation, may hinder reading performance by limiting the ability to read and integrate several words per fixation."

p17. I note that the target words varied quite a lot in terms of length. This will also determine (to some degree) how effectively they were parafoveally processed. This underlines the importance of considering the neural data in relation to the eye movement data in more detail. Were the effects similar for long and short words - I think that the authors must predict that the parafoveal effects would be reduced for long relative to short words, right?

We appreciate the opportunity to clarify this point. While the target words in our study ranged from 3 to 8 letters, their lengths were tightly distributed around the mean (M = 5.1 letters, SD = 1.2, added to the manuscript), with short or long words being rare: only 1.7% of target words were 3 letters, 5% were 7 letters, and 1.7% were 8 letters. The

majority (92.6%) fell within the 4–6 letter range. We intentionally minimised variability in word length to reduce potential confounding effects on parafoveal processing, while also having some shorter and longer words. We do think that parafoveal effects might be reduced for longer compared to shorter ones because short words are more likely to fit within the perceptual span, whereas longer words extend further into the parafovea, where resolution is lower. However, our current dataset lacks sufficient long words to reliably test this hypothesis. We have added these distribution details to the Methods section (p.20, line 405 in the revised manuscript) and added them to the Discussion (p.15, line 294 in the revised manuscript):

“The length of target words ranged from 3–8 letters ($M = 5.1$, $SD = 1.2$), with most words (92.6%) being 4–6 letters long.”

“It should also be noted that the target words used in our study are relatively short (mostly between 4 to 6 letters), the parafoveal effect observed in the current study may be reduced when parafoveal words are relatively longer, which deserves further investigation in future studies.”

P20. The characterisation of the plausibility assessment is inaccurate here. Three types of sentence were adopted: grammatically legal plausible sentences (the experimental sentences); grammatically legal but implausible sentences; grammatically illegal sentences. The latter category cannot be evaluated in respect of their plausibility since they do not have a grammatically legal form. In fact, I cannot understand how the participants could have been required to form a judgment as to the plausibility of word strings with this form - can the authors explain this? I note that these issues were considered in detail in the paper by Rayner et al. (2004).

Rayner, K., Warren, T., Juhasz, B., & Liversedge, S.P., (2004). The Effect of Plausibility on Eye Movements in Reading. *Journal of Experimental Psychology Learning Memory and Cognition* 30(6):1290-301.

We appreciate the reviewer’s comments on our characterisation of the plausibility test. To clarify, all sentences in our plausibility test, including those labelled “implausible,” were grammatically legal (e.g., “Jeremy quenched his thirst with a glass of program” follows English syntax while being semantically nonsensical).

While Rayner et al. (2004) employed different terminology (plausible, implausible, and anomalous), their methodology similarly yielded a range of plausibility ratings (high, medium, and low) across these categories. Our original labels were intended to capture

the continuum of plausibility judgment. We have now revised our terminology to better align with Rayner et al.'s framework and expanded our description of the rating procedure in the Methods section (p.23, line 448 in the revised manuscript):

“The data were collected by Qualtrics using a 7-point rating scale. The test included 360 experimental sentences, along with 200 filler sentences—100 implausible and 100 anomalous—to occupy the full range of the plausibility scale. Participants were instructed to read each sentence and rate its plausibility: (1) if the sentence did not make any sense, (4) if it was unlikely but still possible, and (7) if it was fully acceptable. Before beginning the task, participants were shown example sentences (not included in the actual test) with corresponding ratings and were encouraged to use the full scale. In the example below, the first sentence was from our experimental material, while the second and third sentences were implausible and anomalous, respectively.”

Minor points

p3, Line 36. "Change "to-be-focused" to "to-be-fixated". This is not an issue of accommodation.

Done.

p4, Lines 55 &56. Delete "the" from "the preview benefit". I suggest this change be applied throughout.

Done.

p4, Line 70. This sentence should be rewritten. The effect does not exert any influence over saccadic planning. The effect reflects influences on processing.

Done.

p6, Line 108. Change "fixated upon" to "fixated". Again, I suggest this change be applied throughout.

Done.

P18, Line 344. The analyses should be re-run without the 4 stimuli that differ by two letters, not just one. This is a standard procedure for eye movement experiments.

We have re-run the analyses after excluding the 4 orthographic neighbours differing by two letters. The parafoveal orthographic effect remained statistically significant ($p < .001$, cluster-based permutation test: $p < .001$), with a slightly adjusted time window (66–189 ms, originally 68–186 ms). We've added this to the results part (p.9, line 157):

“Furthermore, to maintain a consistent definition of orthographic neighbours across pairs, we excluded four orthographic-within pairs that differed by two letters rather than one; the parafoveal orthographic effect remained robust (66–189 ms, cluster-based permutation test: $p < .001$).”

p18, Line 351. Change "ensure a same" to "ensure a similar level of".

Done.

P19, Line 376. Even though the predictability levels were quite low, they still sit at 20%. This is actually quite a high value - I'm surprised by this. I suggest that this receives some comment in the text.

We appreciate the reviewer's attention to detail and the opportunity to clarify this point. The original sentence "... and no sentences were highly constrained (the average predictability for the most predicted non-target words was $20.3 \pm 10.4\%$)" refers to whether the sentences were highly constrained—that is, whether a specific word was strongly expected by most participants. The predictability rating here applies to the **most predicted non-target words**, not the actual target words used in the experiment. We've edited the manuscript to make it clearer (p.22, line 441 in the revised manuscript):

“In the final version of sentences, the average predictability of the target words was $0.9 \pm 3.0\%$ (mean \pm SD), indicating that the target words were unpredictable; and the average predictability of the most frequently predicted non-target words was $20.3 \pm 10.4\%$ (mean \pm SD), suggesting that the sentence contexts were not highly constrained.”

A number of the references in the Reference section are not formatted correctly and do not provide full reference information (e.g., 1&2 and 19&20 inconsistently list the same journal; 6, 8, 11, 25, 41, 44 and 45 are incomplete).

We appreciate the reviewer's attention to the accuracy of our references. We have carefully reviewed and corrected all formatting inconsistencies and missing details in the Reference section.

Reviewer #2 (Remarks to the Author):

The authors report MEG evidence to suggest that, during reading, orthographic and semantic information is extracted from parafoveal words around ~70ms and ~140ms, respectively.

Summed up like that in a single sentence, this may not sound like much, but it actually is. This is an excellent way of exploiting MEG: the combination of the preview boundary paradigm with representational similarity analysis (RSA) is clever, and thanks to the high spatio-temporal resolution of MEG produces a very clear picture of the timecourse of parafoveal information extraction. In my view the RSA approach provides an important new angle on a longstanding debate, and therefore I deem this study worthy of publication in Nature Communications after minor revisions. I've added some comments below.

Joshua Snell

We sincerely appreciate the reviewer's positive feedback. We have carefully addressed the revisions to further strengthen the manuscript. Below, we provide detailed responses to each comment.

p.3 line 42: It is claimed that most studies have focused on a single level of word information. That is true. However, in Snell, Meade, et al. (2019, Neuropsychologia) we showed correlates of parafoveal processing both at sub-lexical and lexical levels. Perhaps the authors mean to say that most studies have manipulated only one word dimension (e.g., orthographic flanker overlap in aforementioned study); but that single manipulation may nonetheless inform multiple levels of processing.

We thank the reviewer for this comment, which prompted us to remove the controversial sentence. We do appreciate the findings in Snell et al. (2019), which were cited in the old and revised manuscripts.

p.4 line 58: The authors claim that preview benefit effects do not reveal the timecourse of parafoveal information extraction, but some preview benefit studies have in fact addressed this; see e.g. the 'fast-priming' paradigm of Hohenstein, Laubrock & Kliegl, 2010, in JEP:LMC. The authors may note that the method of Hohenstein et al. had a shortcoming though: they replaced previews at some point during the fixation on the pre-target. These sudden changes in the visual field likely capture attention and thus produce a distorted image of how much parafoveal processing there typically is. Long story short, the timecourse of parafoveal processing is not a new topic; but novel methods are certainly welcome.

We agree that the timing of parafoveal processing is not a new topic, and that preview benefit studies, including Hohenstein et al. (2010), have contributed valuable insights. We have deleted this sentence in the revised manuscript.

p.5 line 74: "Earlier neural mechanisms reflecting the onset of parafoveal semantic processing may exist but have yet to be identified." In Snell et al. (2023, Cortex), we report FRP effects caused by parafoveal grammatical violations as early as 100ms after fixating the preceding word. The authors talk specifically about semantics, but I'd say syntax and semantics go hand in hand: to notice the grammatical violation, readers had to extract some meaning from the parafovea.

We agree that syntactic and semantic parafoveal processing are interconnected, and grammatical violations likely involve preliminary semantic extraction (e.g., word category, argument structure) to detect syntactic anomalies. We acknowledge that our original phrasing ("earlier mechanisms... have yet to be identified") could imply that no earlier studies have found evidence for early parafoveal semantic processing, we have revised the text on page 5, line 81 accordingly:

"Earlier neural mechanisms reflecting the onset of parafoveal semantic processing may exist (Pan et al., 2023; Snell et al., 2023), but the existing evidence remains limited."

At several points in the Intro, the authors highlight the fact that they track orthographic *and* semantic information extraction *from the same word*. Maybe that's not the novelty that needs to be emphasized. Firstly, many preview benefit studies have similarly used various preview conditions to assess various levels of processing in a single experiment. Secondly, it would be equally fine to have several separate studies reporting on different levels of processing; together they provide a complete picture. I think you're OK by just emphasizing the novelty of the RSA approach.

We agree with the reviewer and have toned down the emphasis on tracking orthographic and semantic processing from the same word. For instance, on page 7 line 134 we now use "from the upcoming word" instead of "from the *same* word".

Methods: The authors could justify the use of the boundary paradigm a bit more. For the RSA, you just need contrasts between neighbors and unrelated words - and one might argue that you don't need the boundary paradigm at all for this. Just inspect neural activity during the fixation on 'clever' in

- the clever waiter
- the clever writer
- the clever author
- the clever scouts

see where and when contrasts [waiter-writer] and [waiter-author] are different from [waiter-scouts] and you're done?

I anticipate the justification would be that you want all conditions to be perfectly equal once the eyes leave 'clever', but most of the magic happens before that moment anyway, right?

We would like to clarify that our study did not employ the boundary paradigm. We suspect that certain wording in the manuscript may have inadvertently suggested this, and we apologise for any confusion. We have updated the manuscript throughout to avoid similar confusions. For instance, we replaced “orthographic/semantic previewing effects” with “orthographic/semantic parafoveal effects”.

Results: Although you had significant differences, it is quite striking that, overall, correlations were always really low (Pearson $R < 0.07$). A few words on this would be good.

We appreciate the reviewer’s observation regarding the relatively low correlation values in our RSA results. While the correlations were indeed small, this is in line with previous reports in the literature. First, our analysis involved all 306 MEG sensors, many of which do not contribute to the effect, thereby diluting the overall correlation strength (somehow counter-intuitively, the absolute correlations decrease with the number of sensors, albeit the effects may become more robust). Second, our RSA was performed on single-trial data, which is inherently more variable than averaged data, leading to lower correlations. The R values in our studies are within the typical range reported in MEG-RSA studies, particularly those performing RSA at the single-trial level (e.g., Wang et al., 2018, 2020).

On page 8, line 154 in the Result section, we now write:

“We note that the values of the correlations are relatively low, but they are comparable to previously published studies using similar approaches⁴⁸⁻⁵¹. As the statistical testing demonstrates, the differences in correlations are robust across participants.”

Reviewer #3 (Remarks to the Author):

This is a review of the paper titled “Fast hierarchical processing of orthographic and semantic parafoveal information during natural reading”, submitted for publication in Nature Communications by Lijuan Wang et al..

This is a signed review. I’m Davide Crepaldi (SISSA, Trieste and the University of Pavia), and wrote the following notes with substantial help from my postdoc Giulio Severijnen, in

the context of the (brilliant, in my view) Early Career Researcher co-reviewing initiative of the journal.

Overall, this seems like a nice, solid paper, which brings good theoretical insight and is methodologically sound. The experiment is ingenious, and does provide important information about word processing during natural reading in a way that previous studies couldn't. The paper is also very nicely written – clear, compelling, concise. All in all, neither me nor Giulio see big obstacles to publication in NatComm.

However, we do have a number of theoretical and methodological points that we'd like the authors to take up in a revision – some of which we see as quite major.

We sincerely thank Dr. Davide Crepaldi and Dr. Giulio Severijnen for their thoughtful evaluation of our manuscript and their constructive feedback. We have carefully revised the manuscript to incorporate their feedback, ensuring that both major and minor concerns are thoroughly resolved.

Re: the theoretical framing of the study, I'm not super happy with the authors' characterization of the eye tracking, invisible-boundary findings. They give it for granted that semantic parafoveal effects are well established in that literature, and present them as essentially on par with orthographic and phonological effects (around line 52). I'm not necessarily on top of the latest development in this field (so apologies if I'm missing something crucial here), but my understanding is that while there's wide agreement that orthography is processed in the parafovea, eye tracking experts are still strongly debating the existence of semantic effects.

We acknowledge the ongoing debate regarding semantic parafoveal processing. We've expanded this paragraph to reflect this ongoing debate. Referee 1 made a similar point. On page 4, lines 54–66, we now write:

“Preview benefits have been consistently observed for orthographic^{13–15} and phonological^{16–18} features, supporting parafoveal processing of these aspects. Semantic preview benefits, however, remain debated: earlier studies found no evidence for semantic preview benefits^{19–21}, while later research suggested that semantic preview benefits occur only under specific conditions^{22–26}, such as a capitalised initial letter²² or a constraining context²⁶ of preview/target word. Other eye movement studies have investigated parafoveal processing by measuring which characteristics of parafoveal words influence the processing of the currently fixated word, i.e., parafoveal-on-foveal

(PoF) effects. While orthographic PoF effects are well established (for a review see¹), the results from eye movement studies have largely not provided evidence in favour of lexical and semantic PoF effects^{27,28}. Taken together, eye-tracking evidence remains inconclusive as to whether semantic features can be extracted from the parafovea.”

To my eyes, also the FRP data characterization is a bit questionable. The authors take the fact that FRP semantic effects appear relatively late — later than what would be necessary for that information to influence the upcoming saccades — as reason to believe that there’s an earlier component that has gone unnoticed thus far (around line 70). This logic only stands if one assumes that parafoveal words’ semantics must be part of the processing leading to saccade decisions. The possibility is not considered that this might simply not be true – and therefore, the FRP semantic effect is all there is to observe and is, so to speak, “appropriately late”.

Now, there is nothing objectively wrong in the authors’ approach, and of course the present results do show (and convincingly so) that there was indeed an earlier component. However, the storytelling seems to give much a priori justification for the results that are eventually found – more than the pre-existing literature would justify, I would contend.

To be clear – this does not change my conviction that the results are interesting and solid. I don’t believe that data are epistemologically stronger when they are more strongly expected ex ante. Still, I would personally appreciate a more balanced layout of the landscape for this study.

That’s a very good point. Our original argument—that an earlier semantic processing component may exist—relied on the assumption that parafoveal semantic information contributes to saccade planning. For instance, when a word is skipped in reading, it can be reasonably assumed that it had been sufficiently identified before fixation. However, as the reviewer points out, the empirical link between semantic extraction and saccadic decisions remains unresolved, so we have removed the points regarding saccadic planning. We agree that some of our phrasing might give the impression that an early semantic effect was strongly predicted based on prior literature. We’ve toned down the a priori justification while emphasising the exploratory nature of our study in the Introduction (p.5, line 77 in the revised manuscript).

“However, the parafoveal N400 effect typically occurs over 250 ms after fixation onset on the pre-target word—by which the target word is often already in the fovea, as typical fixation durations during natural reading are ~200 ms. Consequently, the parafoveal

N400 effect likely captures a later stage of parafoveal semantic processing. Earlier neural mechanisms reflecting the onset of parafoveal semantic processing may exist (Pan et al., 2023; Snell et al., 2023), but the existing evidence remains limited.”

Partly related to the previous point, the semantic effects that are found here are really quite early – probably earlier than most of the MEG literature on semantic processing of printed words. To illustrate with an example that I know well :), Vignali et al. (2023) found that the earliest semantic component in an MEG study on isolated printed word arises around 300ms post stimulus onset. In the same study, word frequency – which tracks lexical identification and is therefore widely thought to precede meaning computation – emerged no earlier than 300ms post. Of course, we can see reasons why semantic processing might have emerged earlier in the present work (e.g., sentence embedding allows prediction), but also reasons why it might have emerged later (e.g., foveal processing of isolated words, as in Vignali et al., must be more perceptually powerful than parafoveal processing). Again, this has implications for the rationale of the present experiment (ex ante), and for the discussion of the present results (ex post). The authors discuss quite in depth the connection between the present data and FRP-derived N400 (around line 270); less or no emphasis is given to the integration of the current findings with MEG-derived timelines for visual word processing more generally.

That’s a great point. We agree that our Discussion focused primarily on FRP/N400 comparisons while neglecting integration with MEG-derived timelines for semantic processing more generally. We believe our natural reading paradigm contributes to the earlier semantic effects observed in our study (though likely not because of prediction, as this was kept low). In studies where participants process isolated words, such as Vignali et al., words are processed after fixation. In our natural reading paradigm, words in a sentence are simultaneously presented, allowing one or more words to be processed prior to fixation. We have revised the manuscript to incorporate this discussion (see p.16, line 320 in the revised manuscript).

“The early timing of parafoveal orthographic and semantic processing observed in our study might also be partly explained by the naturalistic reading paradigm we adopted. In MEG studies where participants process isolated words (Vignali et al., 2023), word processing of a target word N occurs after fixating on it. However, in our study, words in a sentence are simultaneously presented, allowing one or more words to be read prior to fixation. Processing of the target word may have been initiated at the N-2 word position (or even earlier), thereby facilitating faster access at word position N-1, a possibility that

is not available in isolated word designs. This would be an important question to explore in future research.”

There's a correlation between the magnitude of the parafoveal effect(s) and reading speed. This is interesting and makes perfect sense. However, correlation isn't causation, and we see plausibility in both causal directions – quicker readers make more use of parafoveal information to support their speed, or readers with better grasp of parafoveal information can then read more quickly. The authors use a wording (e.g., “predict”) that might imply a causal interpretation, and might seem to privilege one causal direction over the other (e.g., lines 323-324). However, this isn't entirely clear; maybe the authors can reformulate more explicitly and/or elaborate more?

We fully agree that correlation does not imply causation, and our original phrasing (e.g., “predicted”) risked implying a unidirectional causal interpretation. We have carefully reworded our statements to avoid implying causation. For instance, on page 19, line 384 in the revised manuscript, we now write:

“This parafoveal processing was associated with individual reading speed, with stronger parafoveal effects observed in faster readers.”

A final theoretical point concerns the content of the orthographic and semantic computations that the authors have uncovered, and their overlap. This was mostly triggered by the authors' explanations for the timing of the effects. They show that the orthographic effect takes place between 68-186 ms and the semantic between 137 - 247 ms. In the discussion (around line 266), they conclude that this timing aligns well with visual information reaching the visual (50 ms) and temporal (70 ms) cortices. While the lower boundary makes sense, we thought it remains a bit unclear what happens *until* 186 ms. What type of orthographic processing is still happening at 186 ms? The authors present in several passages what might seem as a quite serial view of printed word processing. So, we're wondering what they would make of the overlap between the orthographic and the semantic windows.

A similar kind of argument can be made for the semantic effect. The authors state that semantic processing is well aligned with the estimated timing constraints, allowing to affect the decision to skip the target (p.14, line 277). But we don't really know that participants actually do this – effectively use this semantic information to inform saccade decisions.

This is probably where the authors' natural reading paradigm shows its limitations; while it is certainly very ecological and does provide a new look into these theoretical

questions, it's difficult to go deeper in terms of how (and whether, in fact) the information that the brain encodes is then used cognitively. Maybe the authors can find room to discuss this point?

We appreciate the reviewer's insightful comments. The temporal overlap (137–186 ms) between orthographic (68–186 ms) and semantic (137–247 ms) processing suggests that semantic processing (starting at ~137 ms) may start once sufficient orthographic information is available (e.g., partial letter sequences), without needing a complete orthographic resolution. Another possibility is that the overlap might be an intermediate stage of orthographic-to-phonological conversion (e.g., grapheme-phoneme mapping).

We've added a paragraph to discuss this (p.17, line 329 in the revised manuscript):

“Our results demonstrated that the different stages of parafoveal processing are staggered: low-level orthographic information is available first (~68 ms) (Fig. 3a) and higher-level semantic information is available relatively later (~137 ms) (Fig. 3b). Additionally, these processes overlap temporally: semantic processing begins while orthographic processing is reflected in the data until ~186 ms. This suggests that semantic processing may start once sufficient orthographic information (e.g., partial letter sequences) is available, while further orthographic detail is processed. Another possibility is that the overlap might reflect an intermediate stage of orthographic-to-phonological conversion (e.g., grapheme-phoneme mapping), which could co-occur with early semantic activation. The observed temporal overlap between orthographic and semantic processing supports a partially parallel model of word recognition rather than a strictly serial one.”

While our study provides evidence for early parafoveal semantic processing, we agree that we cannot determine whether or how readers actively use it for saccade planning. We've deleted the saccade planning point and revised the discussion as below (see p.16, lines 310–320 in the revised manuscript):

“How does the timing of the N400 (emerging at 250–300 ms) relate to the ~137 ms onset of semantic parafoveal processing found in our study? As we typically saccade every 200–300 ms during natural reading, as we read 3–4 words per second, the parafoveal N400 emerges when the parafoveal word is often already fixated. The E/FRP paradigms may not be sensitive enough to detect an early semantic effect, as they rely on phase-locked, averaged neural responses that may miss transient or subtle dynamics. It is also worth noting that N400-type studies typically capture higher-level processes, such as the integration of the parafoveal target word into sentence context,

whereas our approach isolates the neuronal activity associated with parafoveal semantic information at the single-word level.”

On more methodological grounds, we have the following questions/suggestions.

We wonder whether the authors have addressed the possibility that on some trials, their participants might have skipped ahead within the core of the sentences right when the sentences appeared, to then return at the beginning. I have seen quite a lot of this in my own natural reading studies. Although we wouldn't know what kind of processing these early run-ahead fixations might trigger, this might of course mess up the timing a bit (e.g., participants might have had some early orthographic processing?).

This is an excellent point. We took specific steps to ensure that run-ahead fixations did not influence our analyses. First, only the first fixation on the pre-target word was included. Second, we excluded any trials in which the pre-target word was initially skipped and later fixated. These steps ensure that our neural measures reflect parafoveal processing during the *first encounter* with the pre-target word, avoid contamination from later regressive fixations and potential confounds from delayed or secondary processing unrelated to parafoveal processing. We have revised the relevant section in the Methods (Eye Movements section, p.27, line 534) to make these steps clearer:

“To ensure that our neural measures reflect parafoveal processing upon the first encounter with the pre-target word, we selected only first fixations on the pre-target word during the first pass of reading. If a participant initially skipped the pre-target word and later returned to fixate it, those trials were excluded. These steps eliminate potential contamination from later regressive fixations and potential confounds from delayed or secondary processing unrelated to parafoveal processing. Fixations that were too long (>1000 ms) or too short (<80 ms) were also excluded from the analysis.”

Again on the methods, while most of the details are thoroughly discussed, we wonder about the specifics of the stimuli and the experimental procedure. For example, the authors do provide one example of a sextet (so two triplets that have the same pre-target word), but we seem not to be able to find the entire stimulus list (also not on OSF). It would be nice to have it somewhere.

We thank the reviewer for highlighting this omission. We've now uploaded the full stimulus list to OSF.

Related to this, we wonder if any effects could be explained by the specificity of the triplet pairings. Each triplet was matched with another triplet, after which the data were averaged. This is appropriate of course. However, was there any substantial item variability? Did some triplets show larger effects than others? Are there triplets that don't show the effect? It would be good to get a feel for how much the triplet pairings contributed to variance in the results (over and above the general guarantee given by the statistics).

This is a very good point. As this study represents a novel and exploratory application of RSA to natural reading, we intentionally decreased the lexical variability of pre-target/target words to optimise RSA sensitivity. For example, the lengths of target words were tightly distributed around the mean ($M = 5.1$ letters, $SD = 1.2$), and 92.6% were 4–6 letters. Also, the word frequency mostly fell within a medium-frequency range. As a result, item-level variability was generally low. While this approach successfully revealed robust effects, it limits our ability to assess how effects vary with item-level properties. We have expanded the Discussion (p.15, Line 297 in the revised manuscript) to highlight this limitation and future directions.

“Lastly, our design intentionally minimised the lexical variability of pre-target and target words to optimise the sensitivity of the RSA, but this approach also somewhat limits the insight into how various word-specific factors shape parafoveal processing in natural reading. Future work could extend our approach by incorporating a broader and systematically varied set of materials—modulating factors such as word length, frequency, and predictability—to establish the key factors and boundary conditions for parafoveal processing.”

Within every triplet, the same control sentence is being used to compare orthographic and semantic effects (e.g., O: writer - waiter, C: writer-police; S: writer-author, C: writer-police). Perhaps it would be good to check if there aren't any effects due to choosing one specific control sentence. For example, the authors might run the same analyses, but with a different control sentence (e.g., writer-policy)?

We conducted supplemental analyses using a new set of randomly selected control sentences, distinct from those used in the main analysis. The results remained consistent with our main findings, suggesting that the observed effects are not driven by the specific choice of control words. But we agree that the selection of control words can introduce bias to the RSA result: it is important to ensure that the control word has no

orthographic or semantic overlap with the critical word. We have made this clearer in the Method section (p41, line 412).

“This pairing established orthographic within-pair (e.g., “writer–waiter”) and semantic within-pair relationships (e.g., “writer–author”), while also providing unrelated between-pair controls (e.g., “writer–police” / “writer–policy” / “writer–guards”, randomly chosen in analysis).”

Finally, regarding the experimental procedure more generally, we couldn't find how the experimental lists were randomized. The authors have 360 sentences, each of which is shown once. This is divided into 10 blocks of 36 sentences. Was this arrangement fully random? Did participants see sentences of the same triplet in the same block? And what about the order of the sentences? Did participants for example always see first the target sentence (writer) and then the orthographic (waiter) and semantic one (author)? Or was this randomized, too? The reason we ask is that perhaps there could be some spill-over effects from one sentence to the other. Maybe the neural response (in terms of RSA, but also in terms of the timing of the effects) on waiter/author is different when participants first saw “writer”, compared to when they did not?

We appreciate the reviewer's thorough inquiry into the experimental design. Below, we clarify the randomisation and safeguards against spill-over effects:

First, the order of sentences was not fully random; instead, it was initially randomised and then adjusted to ensure that sentences within a sextet were presented far apart in the experiment. On average, 58 other sentences (minimum 35) appeared between any two sentences from the same sextet, as indicated in the Stimuli part of Methods (p.18, line 354 in the original manuscript). There were 36 sentences in each block, so participants were not presented with sentences of the same sextet in the same block. This separation minimises carryover effects by preventing immediate exposure to related sentences (e.g., "writer," "waiter," "author").

We also added the following to the Methods section of the manuscript (see p.21, line 422 in the revised manuscript):

“Additionally, to control for order effects, the 360 sentences were divided into two halves (180 each). For half of the participants, the second half was presented first, and vice versa.”

Reviewer #4 (Remarks to the Author):

REVIEWERS' COMMENTS

Reviewer #1 (Remarks to the Author):

I would like to thank the authors for considering my comments and for taking the time to revise their manuscript to deal with the points I raised. I think that they have done a very good job in their revisions and responses. I think that the revised manuscript is much stronger and I would like to see it published. Again, I am grateful to the authors for responding to my points.

Simon P. Liversedge

We sincerely thank the reviewer for the generous and encouraging comments. We are pleased that the revisions have addressed the reviewer's concerns.

Reviewer #2 (Remarks to the Author):

I was already quite enthusiastic about the original manuscript. The authors have taken great care to improve the clarity of the paper, and I think they did a good job responding to the points raised by myself as well as the points raised by the other reviewers. I'm happy to recommend publication of the paper in its current form.

Joshua Snell

We thank the reviewer for the positive feedback and kind words. We are very appreciative for the helpful suggestions during the revision process.

Reviewer #3 (Remarks to the Author):

We thank the authors for their responsiveness to our comments. They addressed all our issues satisfactorily and we now fully endorse the publication of this paper.

We'd only have two remaining minor points that we'd like to ask some clarification on.

We thank the reviewers for their constructive final comments and their overall positive evaluation of our revised manuscript. Below, we provide clarification on the two remaining points.

On p.16, the authors state that “The E/FRP paradigms may not be sensitive enough to detect an early semantic effect, as they rely on phase-locked, averaged neural responses that may miss transient or subtle dynamics”. However, it is unclear to us why this is the case. Are the authors suggesting that ERP paradigms are not fit to detect early effects? We understand that the analyses used in ERP studies can make it difficult to detect subtle, oscillatory effects, but how is this related to the timing of ERPs?

We appreciate the opportunity to clarify this point. Our intention was not to suggest that ERP paradigms are incapable of detecting early effects, but rather to emphasise that multivariate approaches, such as RSA, offer greater sensitivity in capturing the earliest emergence of cognitive processing. Traditional univariate ERP analyses depend on averaging neural activity that is strictly phase-locked to stimulus onset, which helps isolate robust components like the N400. However, earlier neural computations that do not produce detectable univariate ERP deflections might be attenuated or entirely missed. By contrast, RSA can detect distributed patterns of activation across channels, even when the underlying signals are relatively weak. This allows them to uncover early processing dynamics that may precede the emergence of observable ERP components. Indeed, prior work has shown that multivariate decoding can reveal earlier differences between conditions than univariate ERP analyses (e.g., Cauchoix et al., 2012; 2014). To be clear, we agree that the parafoveal N400 component reflects semantic processing in the parafovea. However, its relatively late onset (often beyond the average fixation duration) does not indicate the beginning of parafoveal semantic processing.

We have revised the manuscript (p.16, line 314) to clarify this methodological point:

“The E/FRP paradigms may not be optimally sensitive to detecting early semantic effects, as they rely on stimulus-locked, averaged neural responses that are well suited to isolating robust components such as the N400. However, earlier neural computations that do not produce clear univariate ERP deflections may be missed. Indeed, previous studies have demonstrated that multivariate decoding can reveal condition-related differences at earlier latencies than those identified by univariate ERP analyses (e.g., Cauchoix et al., 2012; 2014).”

The authors then continue by stating that “It is also worth noting that N400-type studies typically capture higher-level processes, such as the integration of the parafoveal target word into sentence context, whereas our approach isolates the neuronal activity associated with parafoveal semantic information at the single-word level”. Here again it is unclear to us why this is the case. First, we’re doubting why

ERP studies should necessarily capture higher-level processes. Second, how then is the present study different from previous studies? Since the present study still involves integration into a sentence context.

To clarify, we did not intend to suggest that ERP studies in general reflect higher-level processing. Rather, we were referring specifically to N400-type ERP studies, which typically manipulate the semantic congruency or predictability of a target word in relation to the broader sentence context, and the resulting N400 effects are often interpreted as reflecting higher-level processes such as semantic integration or prediction error. In contrast, our design uses naturalistic sentence context to preserve naturalistic reading, but we focus on word-specific semantic representations *prior* to sentence-level integration. In contrast, although our study also uses naturalistic sentence contexts to maintain ecological validity, our focus is on word-specific semantic representations prior to their integration into the sentence context. The semantic relationships between target words (e.g., *writer–author*) are context-independent, which distinguishes our paradigm from classical N400 paradigms.

We have revised the relevant sentences in the Discussion section (p.16, line 320) as follows:

"It is also worth noting that N400-type studies typically capture higher-level processes, such as the integration of the parafoveal target word into the sentence context, whereas our approach isolates the neuronal activity associated with word-level semantic access prior to integration into broader context. That is, the semantic relationships between target words (e.g., writer–author) are context-independent, which distinguishes our paradigm from classical N400 paradigms."

We don't necessarily need to see a revised version of the manuscript, but we think providing more substantial argumentation for these points would improve the manuscript further. Thank you again for this interesting work!

(I co-reviewed the revised manuscript again with my post-doc Giulio Severijnen.)

Reviewer #4 (Remarks to the Author):

We thank Reviewer #4 for their co-review of the revised manuscript and are grateful for their contribution to the evaluation process.